# Assessment of Mortality Attributable to Air Pollution in the Urban Area of Pisa (Central Italy) Characterized by Multi-Source Exposures

Elisa Bustaffa [1], Marco Redini [2] and Fabrizio Minichilli [1,*]

[1] Institute of Clinical Physiology, National Research Council, Via Moruzzi 1, 56123 Pisa, Italy; elisa.bustaffa@cnr.it

[2] Municipality of Pisa, Via degli Uffizi 1, 56100 Pisa, Italy; m.redini@comune.pisa.it

* Correspondence: fabrizio.minichilli@cnr.it

**Abstract:** Air pollution is one of the main risk factors for human health. The aim of this study was to provide an Integrated Environmental and Health Impact Assessment (IEHIA) tool to estimate the impacts on both environment and human health in Pisa municipality (central Italy). For each pollutant considered ($PM_{2.5}$, $PM_{10}$, and $NO_2$), both Population-Weighted Exposure (PWE) and Attributable Deaths (ADs) were calculated considering the difference between the PWE and the latest air quality guidelines suggested by the World Health Organization. The PWEs were 16.1 μg/m$^3$, 24.9 μg/m$^3$, and 25.9 μg/m$^3$ for $PM_{2.5}$, $PM_{10}$, and $NO_2$, respectively. The ADs from natural causes due to exposure to $PM_{2.5}$, $PM_{10}$, and $NO_2$ were 63, 29, and 51, respectively. The AD distribution was mainly concentrated in urban areas for particulate matter and in urban and suburban areas for $NO_2$. The results highlighted significantly higher levels of air pollution than the reference levels, with a percentage of ADs from natural causes of approximately 6% of the total mortality in Pisa. IEHIA offers support for environmental and health policies and territorial planning. The authors recommend the adoption of prevention measures aimed at mitigating air pollution in critical areas, with a consequent reduction in avoidable mortality.

**Keywords:** air pollution; Integrated Environmental and Health Impact Assessment; attributable deaths; environmental–health policies

## 1. Introduction

Air pollution is recognized as one of the greatest risks to human health [1]. Since 1987, the World Health Organization (WHO) has periodically published health-based air quality guidelines to help governments and civil society reduce both human exposure to air pollution and its adverse effects. The past two decades have seen a marked increase in scientific evidence on the adverse health effects of air pollution, thanks to advances in both air pollution measurement and exposure assessment. New epidemiological studies have documented the adverse health effects of exposure to high levels of air pollution in low- and middle-income countries, and studies in high-income countries with relatively clean air have reported adverse effects at levels much lower than previously studied. Short-/long-term exposures to air pollution, in fact, cause increases in morbidity and excess mortality for many health endpoints [2], although long-term exposures have much more significant impacts on public health, as exposures at very low levels can cause negative effects [3,4]. Chronic exposure to particulate matter (PM) with a diameter of <2.5 microns ($PM_{2.5}$) and ozone is associated with reduced life expectancy, loss of healthy years, and excess mortality from cardiovascular and respiratory diseases [5–9]. Even if evidence is still limited, recent studies have linked $PM_{2.5}$ exposure to adverse birth outcomes [10–12] and to mortality and morbidity from numerous other non-communicable diseases such as diabetes, neurological disorders, and various forms of cancer [13–18]. Considering the many scientific advances

and the global importance of the WHO Air Quality Guidelines, in September 2021 the WHO updated these values (2021-AQGs) [19], reflecting the enormous impact that air pollution has on global health. For this update, the WHO commissioned systematic reviews and meta-analyses on the effects induced by exposure to the main components of air pollution, with the main objective of offering quantitative recommendations based on health considerations for air quality management, expressed in long-term or short-term concentrations of the main air pollutants. These reviews observed that long-term exposure to PM was associated with an increase in mortality from all causes, from cardiovascular and respiratory diseases, and from lung cancer [20], while exposure to nitrogen dioxide ($NO_2$) was associated with an increase in mortality from all causes, from respiratory diseases, and from chronic obstructive pulmonary disease [21]. Short-term exposures to $PM_{2.5}$, PM with a diameter of <10 microns ($PM_{10}$), $NO_2$, and ozone were associated with an increase in mortality from all causes, particularly for $PM_{2.5}$ and $PM_{10}$, and associations with mortality from cardiovascular, respiratory, and cerebrovascular diseases were observed [22], while exposure to other pollutants was associated with an increase in asthma exacerbation [23]. The most important message of this update is that any reduction in outdoor concentrations of key air pollutants generates health benefits for the population involved, even in areas with low concentration levels. The linear exposureresponse relationships defined by the meta-analyses of this update, also considering the lowest concentrations, show that every individual will benefit from cleaner air [20–24]. These results provide a fundamental contribution to clean air policies and regulations at a global level. They are also fundamental for estimating the potential health and economic benefits resulting from policies that aim to reduce exposure to air pollution.

The Integrated Environmental and Health Impact Assessment (IEHIA), through the calculation of deaths attributable to a certain environmental pressure, is a methodology providing immediately readable and understandable results, as it identifies areas with greater exposure. Therefore, it represents a methodological approach that is particularly useful in providing an answer to the need for knowledge of civil society and stakeholders, a tool supporting environmental–health decision-making policies and planning.

In this article, we report a specific case study regarding the application of the IEHIA for the municipality of Pisa (Tuscany region, central Italy), which has been the subject of attention by the municipal administration for years, and studies on its resident population have been carried out regarding risk estimates between exposure to emissions from the incinerator, various industrial plants, and noise, and the onset of pathologies [25,26].

The main objective of this study is to provide an IEHIA tool to estimate Pisa residents' exposure to air pollution and the resulting health impacts attributable to the difference between Pisa air pollution exposure and the 2021-AQGs [19], and to identify the most impacted sub-areas that need mitigation actions. More precisely, we will calculate the number of deaths attributable to the air pollution excess in Pisa compared to the 2021-AQGs. Most studies still use the values suggested by the WHO in 2005 (2005-AQGs) [27], considering one or more pollutants and applying the IEHIA tool only for mortality from natural causes [28–30]. The studies using the 2021-AQGs are conducted either on a single Italian city, evaluating the health impact due to $PM_{2.5}$ exposure [31], or at a global level, considering only $NO_2$ and only mortality from natural causes [32]. Since the existing risk functions, fundamental for the application of the IEHIA tool, are available only for a few pollutants and for limited adverse outcomes, the available scientific literature is rather scarce and heterogeneous. Furthermore, the 2021-AQGs, which are more stringent than the 2005-AQGs, are values determined by scientific evidence, representing minimum values above which significant adverse effects on human health are documented. Therefore, carrying out the analyses using the 2021-AQGs allows us to know the maximum number of deaths that are attributable to air pollution exposure and therefore avoidable. If the 2005-AQGs are instead used, the number of deaths attributable to air pollution exposure would be underestimated. The novelty of this study lies in using the latest values indicated by the WHO as counterfactual, estimating therefore the overall number of avoidable

deaths caused by air pollution exposure and considering all the pollutants and all the risk functions available in the literature, which was updated by the WHO on the occasion of the publication of the latest guidelines, thus providing a global picture of the impact of air pollution in the city of Pisa in terms of attributable deaths. Furthermore, a further novelty of this study lies in the use of this methodology (scientifically consolidated), commissioned for the first time at national level by the municipality itself, as a tool supporting policymakers in identifying strategies to improve urban environments and public health.

## 2. Materials and Methods

### 2.1. Domain and Population under Study

The municipality of Pisa (Tuscany region, Italy) is characterized by an urban center located in the northeast of its territory and by three settlements along the coast (Marina di Pisa, Tirrenia, and Calambrone, Italy) in the southwestern part of the territory, defined by the administrative boundaries where most of the municipal population is concentrated (Figure 1). The population residing in Pisa in the period of 2016–2019 was georeferenced (Figure 1). The determination of the time period was dictated by the fact that it was the same period for which the environmental data were available.

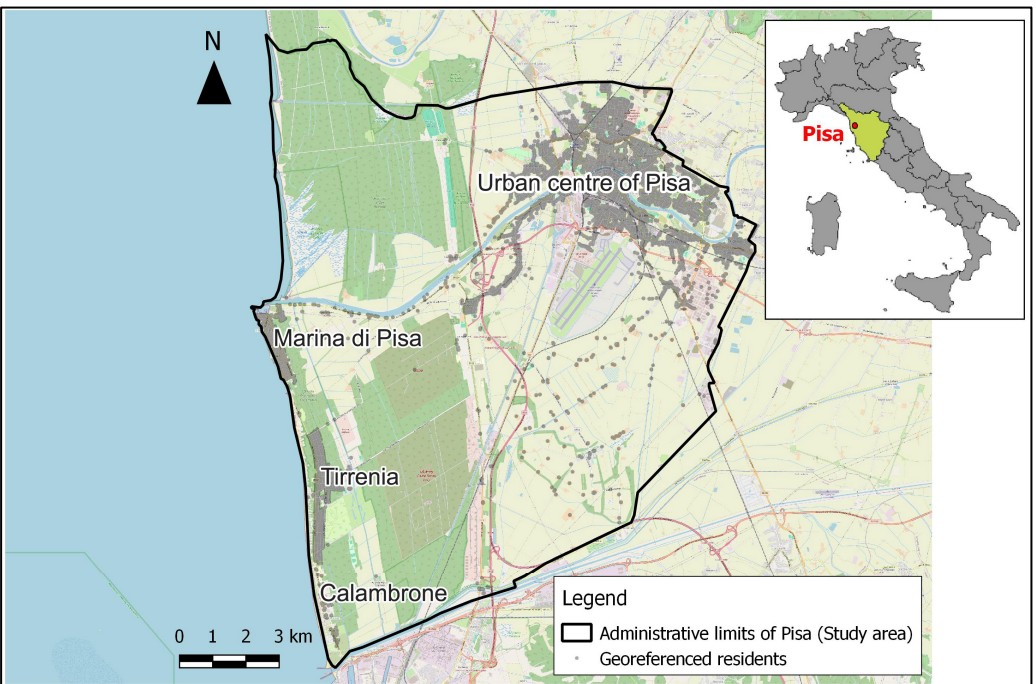

**Figure 1.** Administrative boundaries of the municipality of Pisa and the georeferenced residents. Note—At the top right the map of Italy is reported, with the region of Tuscany in green and the city of Pisa in red.

Pisa has a predominance of tertiary and commercial activities, with three universities, a large research center, a regional hospital, and a complex road network. The study area includes the major sources of pollution (such as main roads and industrial plants, Figure 2) and for which environmental data produced by the regional agency for environmental protection are available. Pisa does not have a variant, so the main arteries intertwine within the city and most of the traffic flows into the city itself (Figure 2a). The railway itself crosses the city, and the airport (national and international) is very close to the town center, as highlighted in the grey part at the bottom left of Figure 2b. Furthermore, industrial settlements have shrunk in recent decades; currently, there are only small factories, together with a medium-sized one producing glass. The main industrial, artisanal, and commercial

areas are colored in pink and blue in Figure 2b and are therefore mainly distributed outside of the historic center in the western and southwestern parts of the city of Pisa.

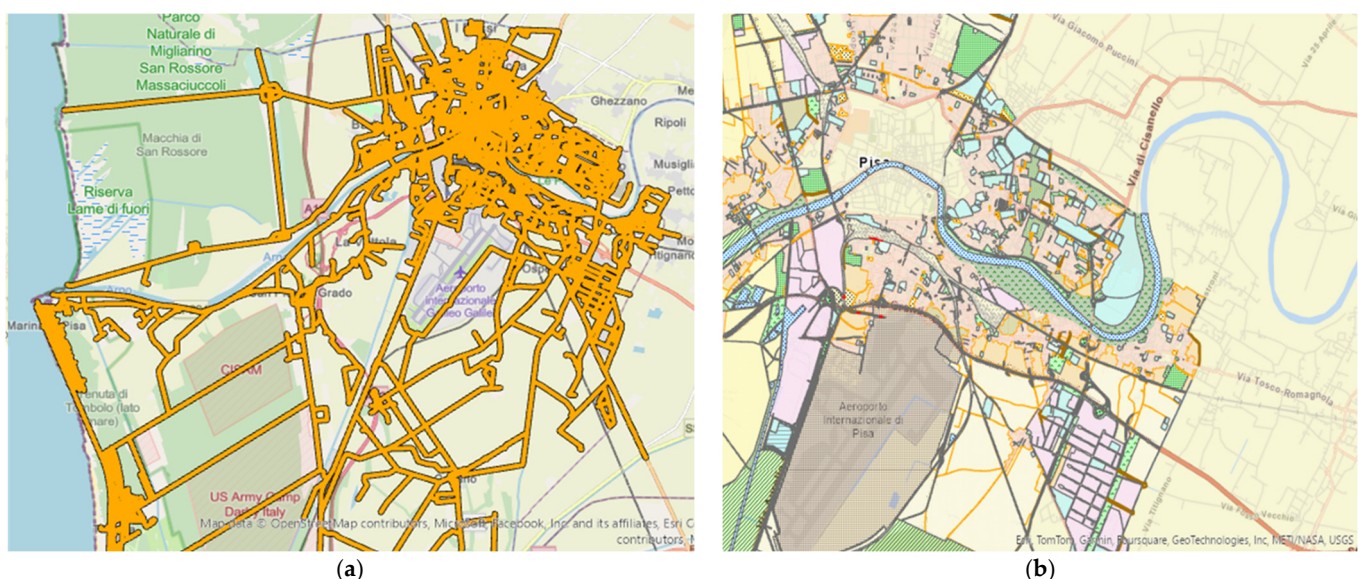

(**a**)                                    (**b**)

**Figure 2.** Road network of Pisa in orange (**a**) and distribution of the main industrial, artisanal, and commercial areas (in pink and blue) and the airport (grey) present in the municipal territory of Pisa (**b**).

*2.2. Exposure Assessment*

2.2.1. Air Pollutant Concentration Maps and Reference Legislation

Air pollution in urban areas can originate from a variety of sources, including industrial emissions, vehicle exhaust, domestic heating, and biomass burning. Air pollutants considered to be priorities given their effects on human health and the extent of their emissions are inorganic gases (sulfur dioxide, nitrogen oxides, $NO_2$, carbon monoxide, ozone), volatile organic compounds (such as benzene and formaldehyde), and PM [33]. A study by Stafoggia et al. (2020) [34] assessed air pollution from different sources by creating pollutant concentration maps for $PM_{2.5}$, $PM_{10}$, $NO_2$, and ozone in the period of 2016–2019. These maps covered the national territory and were obtained by estimating the values of the daily mean concentrations of $PM_{2.5}$, $PM_{10}$, $NO_2$, and ozone using a machine learning approach with random forest methodology. This methodology integrates meteorological, land use, and monitoring data obtained from all available sites provided by the Higher Institute for Environmental Research and Protection together with satellite data on the optical depth of aerosols [34]. Therefore, considering (i) the characterization of the municipality of Pisa (see Section 2.1) (presence of domestic heating, fuel combustion, traffic, and industrial activities), (ii) the availability of the concentration maps, and (iii) the medium–long term adverse effects that some of these pollutants induce on human health, specifically on the cardio-respiratory system, $PM_{2.5}$, $PM_{10}$, and $NO_2$ were chosen as representative of the multi-source exposures in the area under study. In fact, the main source of $PM_{2.5}$ is the combustion of solid fuels for domestic heating, industrial activities, and road transport. As with $PM_{10}$, they can also come from natural sources and can form in the atmosphere. The leading source of $NO_2$ is road transport, which emits $NO_2$ close to the ground, mostly in densely populated areas, contributing to population exposure. Other important sources are combustion processes in industry and energy supply [35].

In this study, the maps by Stafoggia et al. (2020) [34] covering only the area of Pisa municipality, divided into a grid of 277 cells of 1 km × 1 km, were used. From the estimates of the annual mean values of $PM_{2.5}$, $PM_{10}$, and $NO_2$, the means for the available period were calculated and the 2021-AQGs were used for reference values. Table 1 summarizes the 2021-AQGs for the pollutants considered and reports the comparison with both the 2005-AQGs and the Italian legislation. The 2021-AQGs recommend achieving annual mean

values for $PM_{2.5}$ and $PM_{10}$ of no more than 5 μg/m$^3$ and 15 μg/m$^3$, respectively, and no more than 10 μg/m$^3$ for $NO_2$ [19]. The respective values of the 2005-AQGs for $PM_{2.5}$, $PM_{10}$, and $NO_2$ were 10 μg/m$^3$, 20 μg/m$^3$, and 40 μg/m$^3$, respectively [27].

**Table 1.** Latest air quality guidelines recommended by the World Health Organization in 2021 (2021-AQGs) for the pollutants considered compared with both the previous 2005 guidelines (2005-AQGs) and the Italian legislation.

| Pollutant | Time Reference | Ad Interim Target (μg/m$^3$) | | | | AQGs (μg/m$^3$) | | Italian Legislative Decree n. 155/2010 (μg/m$^3$) |
|---|---|---|---|---|---|---|---|---|
| | | **1** | **2** | **3** | **4** | **2021** | **2005** | |
| $PM_{2.5}$ | Annual | 35 | 25 | 15 | 10 | 5 | 10 | 25 |
| | 24 h | 75 | 50 | 37.5 | 25 | 15 | 25 | - |
| $PM_{10}$ | Annual | 70 | 50 | 30 | 20 | 15 | 20 | 40 |
| | 24 h | 150 | 100 | 75 | 50 | 45 | 50 | 50 |
| $NO_2$ | Annual | 40 | 30 | 20 | - | 10 | 40 | 40 |
| | 24 h | 120 | 50 | | | | | |

Legend—$PM_{2.5}$: particulate matter with a diameter of <2.5 microns; $PM_{10}$: particulate matter with a diameter of <10 microns; $NO_2$: nitrogen dioxide; AQGs: air quality guidelines.

Table 2 shows the comparison among the average, minimum, and maximum values of pollutant distribution in Pisa for the period of 2016–2019 and the values of Tuscany, the Po Valley (one of the most polluted areas in Europe), and Italy.

**Table 2.** Comparison among the distribution (mean, minimum, and maximum) of pollutants of interest in Pisa and Tuscany, the Po Valley, and Italy. Period of 2016–2019. Sources [33,36].

| Pollutant | Indicator (μg/m$^3$) Period of 2016–2019 | Area | | | |
|---|---|---|---|---|---|
| | | **Pisa** | **Tuscany** | **Italy** | **Po Valley \*** |
| $PM_{2.5}$ | Mean | 13.6 | 14.4 | 15.7 | 20.6 |
| | Min. | 11.3 | 9.0 | 2.2 | 1.5 |
| | Max. | 17.5 | 23.0 | 68.8 | 96.3 |
| $PM_{10}$ | Mean | 21.3 | 21.1 | 24.0 | 29.1 |
| | Min. | 18.9 | 0.0 | 3.9 | 3.2 |
| | Max. | 26.9 | 31.0 | 102.4 | 117.8 |
| $NO_2$ | Mean | 14.8 | 22.3 | 22.2 | 26.7 |
| | Min. | 7.9 | 2.0 | 0.8 | 1.2 |
| | Max. | 34.0 | 65.0 | 113.1 | 119.2 |

Note—\*: The Po Valley includes the provinces of Milano, Lodi, Pavia, Cremona, Brescia, Mantova, Varese, Bergamo, Piacenza, Parma, Modena, Bologna, Ravenna, Forlì-Cesena, Rimini, Reggio nell'Emilia, Rovigo, Ferrara, and Verona.

In Pisa, the distribution of $PM_{2.5}$ is characterized by a mean value of 13.6 μg/m$^3$, and a minimum and a maximum of 11.3 μg/m$^3$ and 17.5 μg/m$^3$, respectively (Table 2). The distribution of $PM_{10}$ is characterized by a mean value of 21.3 μg/m$^3$, and a minimum and a maximum of 18.9 μg/m$^3$ and 26.9 μg/m$^3$, respectively (Table 2). The $NO_2$ distribution is characterized by a mean value of 14.8 μg/m$^3$, and a minimum and a maximum of 7.9 μg/m$^3$ and 34.0 μg/m$^3$, respectively (Table 2).

For all the pollutants considered, Pisa is characterized by lower mean and maximum values than those of Tuscany, Italy, and the Po Valley; the latter, however, are always higher than the national ones. Only for $PM_{10}$ is the average value observed in Pisa slightly higher than the regional one (Table 2).

Figures 3–5 report, for the period of 2016–2019, the spatial distributions of the average concentrations of $PM_{2.5}$, $PM_{10}$, and $NO_2$ in the study area, divided into 1 km × 1 km cells.

From Figures 3 and 4, we can observe that, for both $PM_{2.5}$ and $PM_{10}$, all the areas of distribution show values higher than the 2021-AQG values of 5 µg/m$^3$ and 15 µg/m$^3$, respectively. A total of 80% of the distribution areas have $NO_2$ values above the limit of 10 µg/m$^3$ (Figure 5). The area with the highest values of $PM_{2.5}$, $PM_{10}$, and $NO_2$ concentrations is the one that covers the urban and suburban center (Figures 3–5).

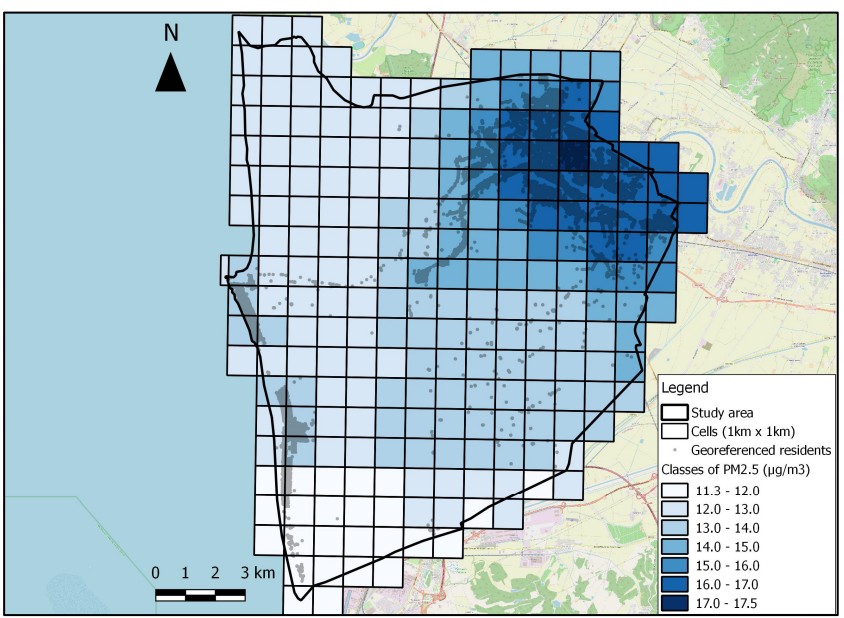

**Figure 3.** Map of the spatial distribution of the average concentration of the particulate matter with a diameter of <2.5 micron ($PM_{2.5}$) for the period of 2016–2019 in the study area of the municipality of Pisa, divided into 277 cells of 1 km × 1 km. Note—$PM_{2.5}$ classes were defined through R's pretty algorithm.

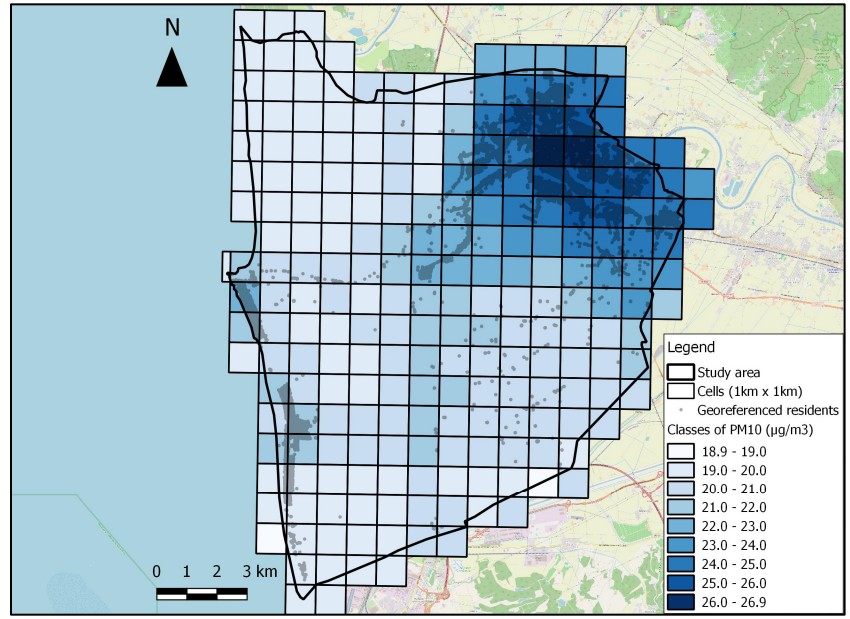

**Figure 4.** Map of the spatial distribution of the average concentration of the particulate matter with a diameter of <10 micron ($PM_{10}$) for the period of 2016–2019 in the study area of the municipality of Pisa, divided into 277 cells of 1 km × 1 km. Note—$PM_{2.5}$ classes were defined through R's pretty algorithm.

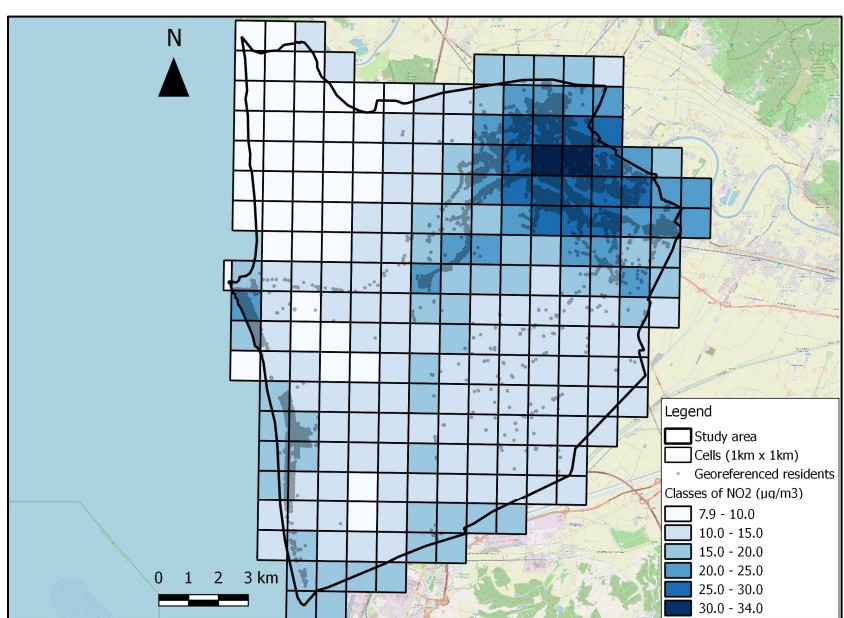

**Figure 5.** Map of the spatial distribution of the average concentration of nitrogen dioxide (NO$_2$) for the period of 2016–2019 in the study area of the municipality of Pisa, divided into 277 cells of 1 km × 1 km. Note—PM$_{2.5}$ classes were defined through R's pretty algorithm.

### 2.2.2. Population-Weighted Exposure

Starting from the georeferenced annual populations, for each cell and for each year under study, the number of residents was counted. Subsequently, for each cell, the average populations in the period considered (AP$_i$ in the following formula, where $i$ varies from 1 to $n$ = 227, i.e., the number of cells) were calculated.

Figure 6 shows the distribution of the average populations per cell of 1 km × 1 km (grid of Stafoggia et al.'s (2020) [34] model) into which the municipal territory is divided. The distribution of the average populations had a mean of 404 subjects per cell, a minimum of 0, and a maximum of 7905 subjects, and the average population for the entire area of study ($AP = \sum_{i=1}^{n} AP_i$) was 91,761 subjects.

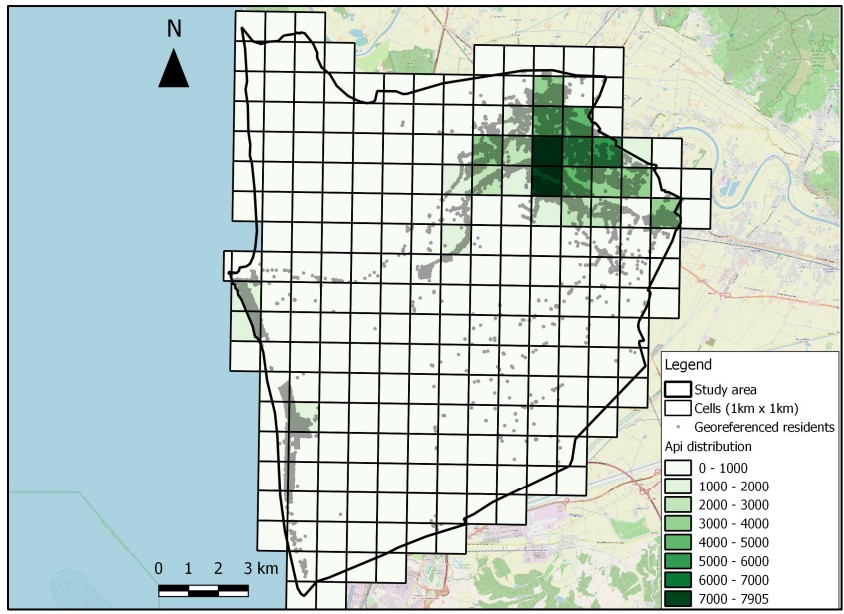

**Figure 6.** Distribution of average population (AP$_i$) considering the grid of the maps by Stafoggia et al. (2020) [34].

For each pollutant, the study area's population-weighted exposure (*PWE* in the following formula) was calculated using the following formula:

$$PWE = \frac{\sum_{i=1}^{n}(AP_i * Concentration\_of\_pollutant_i)_i}{AP} \tag{1}$$

where *i* varies from 1 to *n* = 227, i.e., the number of cells in the territory of the municipality of Pisa.

### 2.3. Calculation of Attributable Deaths

The calculation of the average annual deaths attributable to air pollution exposure was carried out for each 1 km × 1 km cell and for the entire area. To calculate the deaths attributable to the air pollution exposure of a population, it is necessary to have the risk functions, i.e., the concentration–response functions, for each pollutant representative of air pollution and for each cause of death attributable to that specific pollution. The concentration–response functions are expressed as relative risks and correlate the response, in terms of mortality percentage increase, to the concentration of a specific pollutant, usually for an increase of 10 μg/m$^3$ of that specific pollutant.

The causes of death that the scientific literature relates to the pollutants chosen as representative of the air pollution in the territory of Pisa municipality (PM$_{2.5}$, PM$_{10}$, and NO$_2$) are natural causes; cardiovascular, ischemic, and respiratory diseases; and lung cancer.

For each cell and for the entire study area, the attributable deaths were calculated considering the differential (Δ) of the concentrations of the target pollutants compared to the 2021-AQG values, specifically 5 μg/m$^3$, 15 μg/m$^3$, and 10 μg/m$^3$ for PM$_{2.5}$, PM$_{10}$, and NO$_2$, respectively.

The calculation of the attributable deaths (AD in the following formula) was carried out by applying a formula validated and widely used at both the national and the international level [37–40]:

$$AD = A * B * \left(\frac{\Delta C}{10}\right) * AP \tag{2}$$

where:

- *A* is the risk excess in the exposed population, attributable to air pollution. It represents the proportion of the health effect attributable to air pollution and is calculated as

  A = (RR−1)/RR

  where RR is the concentrationresponse function, i.e., the relative risk, derived from the literature, available for the given health outcome. It is generally expressed as relative risk for an increase of 10 μg/m$^3$, as previously indicated.

- *B* is the background mortality rate of the health outcome considered, i.e., the rate that would be observed in the absence of exposure. B is not directly measurable but can be estimated using the following formula:

  $$B = B_0 / \left[1 + \left(\frac{RR - 1}{RR}\right) * \left(\frac{\Delta C}{10}\right)\right]$$

  where:
  $B_0$ is the measured mortality rate of the health effect, referring to the observed concentration, obtained from the available statistical data of the reference population.

- ΔC/10 is the concentration variation for which the effect is to be assessed; it represents the difference between the concentration of the pollutant per cell or for the entire area (using population-weighted exposure as previously calculated) and the reference concentration (counterfactual). This concentration value is divided by 10 since the relative risk, as indicated above, is conventionally expressed in increments of 10 μg/m$^3$.

- *AP* is the average population exposed for each cell or for the entire area, calculated as previously reported in Section 2.2.2.

To calculate "A", it is necessary to know the relative risk values for each health outcome derived from concentration–response functions for exposure to the pollutants of interest. In the latest update of the air quality guidelines [19], the WHO reports the updated risk functions for $PM_{2.5}$, $PM_{10}$, and $NO_2$, and for the pathologies considered to be related to these pollutants (Table 3). Results will also be reported using the percentage of deaths attributable to exposure to the pollutants under study compared to the total average number of deaths in the period of 2016–2019 in the municipality of Pisa.

**Table 3.** Relative risks derived from concentration–response functions for $PM_{2.5}$, $PM_{10}$, and $NO_2$, updated by the World Health Organization in 2021. Source [19].

| Increase of 10 μg/m³ | Causes of Death | Concentration–Response Function (Relative Risk) | CI (95%) |
|---|---|---|---|
| PM$_{2.5}$ | Natural causes | 1.08 | 1.06–1.09 |
| | Cardiovascular diseases | 1.11 | 1.09–1.14 |
| | Respiratory diseases | 1.10 | 1.03–1.18 |
| | Lung cancer | 1.12 | 1.07–1.16 |
| PM$_{10}$ | Natural causes | 1.04 | 1.03–1.06 |
| | Ischemic diseases | 1.06 | 1.01–1.10 |
| | Respiratory diseases | 1.12 | 1.06–1.19 |
| | Lung cancer | 1.08 | 1.04–1.13 |
| NO$_2$ | Natural causes | 1.02 | 1.01–1.04 |
| | Respiratory diseases | 1.03 | 1.01–1.05 |

Note—CI (95%): confidence interval at 95% of probability.

Table 4 reports the age-standardized population mortality rates (regional reference—period of 2016–2019) based on the cause of death of interest; this value represents the $B_0$ component of Formula (2) for calculating the attributable deaths.

**Table 4.** Age-standardized mortality rates for selected causes of death. Regional reference—period of 2016–2019.

| Cause of Death | Age-Standardized Mortality Rate ($\times 100,000$) Regional Reference (Period 2016–2019) |
|---|---|
| Natural causes | 843.53 |
| Cardiovascular diseases | 251.38 |
| Ischemic diseases | 62.73 |
| Respiratory diseases | 62.83 |
| Lung cancer | 46.06 |

## 3. Results

### 3.1. Calculation of Population-Weighted Exposure

The population-weighted exposures, calculated through Formula (1), for each pollutant considered were 16.1 μg/m³, 24.9 μg/m³, and 25.9 μg/m³ for $PM_{2.5}$, $PM_{10}$, and $NO_2$, respectively (Table 5).

**Table 5.** Population-weighted exposures for each pollutant considered and variation between the population-weighted exposures and the guideline values indicated by the World Health Organization (2012-AQG) for each pollutant.

| Pollutant | 2021-AQGs | Population-Weighted Exposure | Δ |
|---|---|---|---|
| PM$_{2.5}$ | 5 μg/m³ | 16.1 μg/m³ | 11.1 |
| PM$_{10}$ | 15 μg/m³ | 24.9 μg/m³ | 9.9 |
| NO$_2$ | 10 μg/m³ | 25.9 μg/m³ | 15.9 |

Legend—2021-AQG: air quality guidelines recommended by the WHO in 2021; Δ: variation between population-weighted exposure and the 2021-AQGs for each pollutant.

The variations in the concentration for each pollutant considered were 11.1 μg/m$^3$, 9.9 μg/m$^3$, and 15.9 μg/m$^3$ for PM$_{2.5}$, PM$_{10}$, and NO$_2$, respectively (Table 5).

The variation in the concentration indicated that population-weighted exposures were significantly higher than 2021-AQG values of 5 μg/m$^3$, 10 μg/m$^3$, and 15 μg/m$^3$.

*3.2. Attributable Deaths*

Applying Formula (2), considering Pisa municipality and the period of 2016–2019, deaths attributable to the difference between the calculated population-weighted exposure and the concentration values of the pollutants of interest (using as counterfactual the 2021-AQG values) were obtained for each cause of death and for each pollutant considered (Table 6).

**Table 6.** Mortality and percentage of mortality attributable to the delta of the target pollutants for the selected causes of death (using as counterfactuals the 2021-AQG values).

| Cause of Death | Pollutant * | Annual Average of Attributable Deaths (CI95%) ** | AD% (CI95%) |
|---|---|---|---|
| Natural causes | PM$_{2.5}$ | 63.0 (48.3–70.2) | 6.1 (4.6–6.7) |
| | PM$_{10}$ | 29.4 (22.3–43.3) | 2.8 (2.1–4.2) |
| | NO$_2$ | 50.6 (22.7–76.5) | 4.9 (2.2–7.3) |
| Cardiovascular diseases | PM$_{2.5}$ | 25.1 (20.9–31.0) | 7.1 (5.9–8.7) |
| Ischemic diseases | PM$_{10}$ | 3.2 (0.6–5.2) | 3.1 (0.5–5.0) |
| Respiratory diseases | PM$_{2.5}$ | 5.8 (1.9–9.6) | 6.6 (2.1–11.1) |
| | PM$_{10}$ | 6.1 (3.2–9.1) | 7.0 (3.7–10.5) |
| | NO$_2$ | 2.6 (0.9–4.2) | 3.0 (1.0–4.9) |
| Lung cancer | PM$_{2.5}$ | 5.0 (3.0–6.4) | 8.4 (5.2–10.8) |
| | PM$_{10}$ | 3.1 (1.6–4.8) | 5.2 (2.7–8.1) |

Legend—*: The guideline values indicated by the World Health Organization in 2021 are 5 μg/m$^3$, 15 μg/m$^3$, and 10 μg/m$^3$ for PM$_{2.5}$, PM$_{10}$, and NO$_2$, respectively; CI95%: confidence interval at 95% of probability; **: to calculate the lower and upper limits of this CI95%, the confidence interval limits of the concentration-response functions were used; AD%: percentage of deaths attributable to exposure to the pollutants under study compared to the total average number of deaths in the period of 2016–2019 in the municipality of Pisa.

Generally, it was observed that the deaths attributable to PM$_{2.5}$ exposure were almost always the highest compared with other pollutants, except for respiratory diseases, for which the highest value of average annual deaths was attributed to PM$_{10}$ exposure.

*Natural causes*—During the period of 2016–2019, 63.0 (CI95% 48.3–70.2) average annual deaths from natural causes attributable to PM$_{2.5}$ exposure were estimated in the territory of the municipality of Pisa (6% of the total mortality from natural causes observed in Pisa), which is the highest number compared to the average annual deaths attributable to PM$_{10}$ (29.4; CI95% 22.3–43.3) and NO$_2$ (50.6; CI95% 22.7–76.5) exposures (Table 6).

*Cardiovascular diseases*—During the period of 2016–2019, an average of 25.1 (CI95% 20.9–31.0) annual deaths from cardiovascular diseases attributable to PM$_{2.5}$ exposure were estimated within the municipality of Pisa (7% on the total mortality) (Table 6).

*Ischemic diseases*—During the period of 2016–2019, within the municipal territory of Pisa, an average of 3.2 (CI95% 0.6–5.2) annual deaths from ischemic diseases attributable to PM$_{10}$ exposure were estimated (3.1% on the total mortality) (Table 6).

*Respiratory diseases*—For respiratory diseases, during the period of 2016–2019, the highest estimated value of 6.1 (CI95% 3.2–9.1) average annual deaths was attributed to PM$_{10}$ exposure (7% of the total number of deaths), while a value of 5.8 (CI95% 1.9–9.6) (6.6%) was estimated as attributable to PM$_{2.5}$ exposure and 2.6 (CI95% 0.9–4.2) (3%) to NO$_2$ exposure (Table 6).

*Lung cancer*—During the study period, average annual deaths from lung cancer of 5 (CI95% 3.0–6.4) (8.4% of the total number of deaths) and 3.1 (CI95% 1.6–4.8) (5.2%) were estimated as attributable to PM$_{2.5}$ and PM$_{10}$ exposure, respectively (Table 6).

In general, of the 63 deaths from natural causes attributable to exposure to $PM_{2.5}$, 25 (40% of the deaths attributable to total natural causes) were due to cardiovascular diseases, 6 (10%) to respiratory diseases, and 5 (8%) to lung cancer. Of the 29 deaths from natural causes attributable to $PM_{10}$ exposure, 3 (10%) were due to ischemic diseases, 6 (21%) to respiratory diseases, and 3 (10%) to lung cancer. Finally, of the 51 deaths from natural causes attributable to $NO_2$ exposure, 3 (6%) were due to respiratory diseases.

Appendix A reports (i) the mortality and percentage of mortality attributable to the delta of the target pollutants for the selected causes of death using the 2005-AQG values as counterfactual (Table A1), which are used in the discussion to compare Pisa's results with other studies, and (ii) the mortality distribution per cell for each cause and for each pollutant considered, accompanied by comments (Figures A1–A5). Overall, for each pollutant and for each cause, a mortality distribution per cell, and therefore a greater health impact, in correspondence with the cells of the urban center, can be observed (Figures A1–A5). The only exception is related to respiratory diseases for $NO_2$, where mortality per cell is distributed not only in correspondence with the urban center but also in the areas of the airport and the coastal zone (Figure A4c). All the analyses were performed using QGis (QGIS desktop 2.18.13. QGIS Geographic Information System. QGIS Association. http://www.qgis.org) and STATA v.15 (Stata Corp., College Station, TX, USA, 2017).

## 4. Discussion

The municipality of Pisa is characterized by several sources of air pollution, including an international airport, industrial/artisanal activities close to residential areas, and a complex road network with a lack of a ring road around the city center. In this situation, most of the air pollution is concentrated in the urbanized area, which suffers from higher exposure.

The diffusion models by Stafoggia et al. (2020) [34] allowed for an estimation of global air pollution from different sources through the mapping of the target pollutants considered in the literature to be factors associated with the risk of mortality from natural causes; cardiovascular, ischemic, and respiratory diseases; and lung cancer [19], namely, $PM_{2.5}$, $PM_{10}$, and $NO_2$. These pollutants were chosen as representative of the air pollution in Pisa due both to the solid evidence on their adverse effects on human health even at low concentrations [19] and to the availability of a consolidated methodology for calculating deaths attributable to exposure to these substances. The main goal of this study is not exactly to "measure" the individual citizens who have died due to air pollution but rather to estimate the phenomenon in the territorial context of Pisa municipality, trying to provide indications on the magnitude of the impact of $PM_{2.5}$, $PM_{10}$, and $NO_2$ air pollution. The reference values used are the 2021-AQGs, which therefore are values derived from scientific evidence above which adverse effects on human health occur, and for this reason, they have an exclusively health-related value. Applying this concept to the data of this study means that the number of deaths from natural causes in the four-year period of 2016–2019 attributable to $PM_{2.5}$, $PM_{10}$, and $NO_2$ air pollution represents the average number of premature deaths that could be annually avoided if the 2021-AQG values of 5 μg/m$^3$, 10 μg/m$^3$, and 15 μg/m$^3$ for $PM_{2.5}$, $PM_{10}$, and $NO_2$, respectively, are reached.

The population-weighted exposures for each pollutant considered were 16.1 μg/m$^3$, 24.9 μg/m$^3$, and 25.9 μg/m$^3$ for $PM_{2.5}$, $PM_{10}$, and $NO_2$, respectively; these are significantly higher than 2021-AQGs of 5 μg/m$^3$, 10 μg/m$^3$, and 15 μg/m$^3$. The estimated value of the percentage of deaths from natural causes attributable to $PM_{2.5}$ exposure was 6.1%, which can reach 6.7% in the worst case (upper limit of the confidence interval); in the best case, the percentage was 4.6% (lower limit of the confidence interval). The estimated value of the percentage of deaths from natural causes attributable to $PM_{10}$ exposure was 2.8%, which can reach 4.2% in the worst case (upper limit of the confidence interval); in the best case, the percentage was 2.1% (lower limit of the confidence interval). The estimated value of the percentage of deaths from natural causes attributable to $NO_2$ exposure was 4.9%, which can reach 7.3% in the worst case (upper limit of the confidence interval); in the best

case, the percentage was 2.2% (lower limit of the confidence interval). Upon analyzing the monitoring data of the Pisa's territory, it was noticed that there is a slight reduction in the concentrations of the pollutants considered, as can be seen from the latest report of the Regional Agency for Environmental Protection [36].

These results highlight the need to keep on working to reduce resident population exposure (in fact, achieving the target values established by the WHO is a challenge). In this regard, the WHO proposes ad interim targets for the various pollutants [19]. Specifically, for $PM_{2.5}$, the second ad interim target corresponds to the limit value defined by European and Italian legislation (25 $\mu g/m^3$), the third is equal to 15 $\mu g/m^3$, while the fourth corresponds to the 2005-AQG value of 10 $\mu g/m^3$. For $PM_{10}$, the second ad interim target is equal to 50 $\mu g/m^3$, which is higher than the limit value defined by European and Italian legislation (40 $\mu g/m^3$); the third is equal to 30 $\mu g/m^3$; and again, the fourth corresponds to the 2005-AQG value of 20 $\mu g/m^3$. The WHO defines these intermediate levels, which are less ambitious than those for which we should aim, precisely to provide achievable objectives to the most polluted countries, in order to motivate them in the development of pollution reduction policies that can be achieved in realistic timeframes [19]. Gradual progress in achieving the intermediate objectives is therefore feasible and desirable.

To perform the usual comparisons with literature data, additional analyses using the 2005-AQGs (10 $\mu g/m^3$, 20 $\mu g/m^3$, and 40 $\mu g/m^3$ for $PM_{2.5}$, $PM_{10}$, and $NO_2$, respectively) were carried out (results reported in Appendix A, Table A1). The estimated value of the percentage of deaths from natural causes attributable to $PM_{2.5}$ exposure was 3.4%, which is lower than what was reported in the Global Burden of Disease published in 2015 [41], which estimated that about 7.6% of total deaths were attributable to long-term exposure to $PM_{2.5}$. Considering the studies carried out in Italy and/or Europe, the one by Khomenko et al. (2021) calculated, in 1000 European cities in 2015, the percentage of deaths from natural causes attributable to $PM_{2.5}$ and $NO_2$ exposure, using as counterfactual the 2005-AQGs [28]. For $PM_{2.5}$, two cities of the Po Valley were among the top 10 cities with the highest preventable number of deaths, with Brescia (population-weighted exposure of 27.5 $\mu g/m^3$) and Bergamo (population-weighted exposure of 26.1 $\mu g/m^3$) in first and second place [28]. More specifically, Brescia and Bergamo showed a preventable annual mortality of 11% and 10%, respectively [28]. For $PM_{2.5}$, the population-weighted exposure for Pisa was 16.1 $\mu g/m^3$ and the percentage of attributable deaths was 3.4%, which is much lower than those of the other two cities that are part of the Po Valley, where air pollution due to $PM_{10}$ and $PM_{2.5}$ is more significant. In fact, in northern Italy, which includes the Po Valley, the high concentrations of $PM_{2.5}$ are due to the combination of a high density of anthropogenic emissions and also meteorological and geographical conditions that favor the accumulation of air pollutants in the atmosphere and the formation of secondary particles [35]. As regards $NO_2$, among the cities with the highest values were Turin and Milan in third and fifth place, with population-weighted exposures of 40.8 $\mu g/m^3$ and 38 $\mu g/m^3$, respectively, while Pisa showed a population-weighted exposure of 25.9 $\mu g/m^3$. For both Turin and Milan, the percentage of deaths from natural causes attributable to $NO_2$ exposure was 0.3 [28], while Pisa showed a percentage of attributable deaths equal to −4.9% (this result of below zero was due to a population-weighted exposure that was lower than the reference value of 40 $\mu g/m^3$). A study conducted in France, in Paris, estimated the number of premature deaths among the population, using as counterfactual the 2005-AQG values. The percentages of estimated attributable deaths were 6.5%, 7.8%, and approximately 5% for $PM_{2.5}$, $PM_{10}$, and $NO_2$, respectively [29]; these results were higher than Pisa's results (3.4%, 1.4%, and −4.9% for $PM_{2.5}$, $PM_{10}$, and $NO_2$, respectively). A study conducted in Texas found a percentage of deaths from natural causes attributable to $PM_{2.5}$ exposure of 0.9% [30], which is lower than what was reported for Pisa.

All previous studies evaluated the percentages of deaths attributable to exposure to certain pollutants just for natural causes and used as counterfactual the 2005-AQG values. Instead, an Italian study estimated the percentages of deaths attributable to $PM_{2.5}$ exposure not only from natural causes but also for all causes for which the literature provides a risk

function, using as counterfactual the 2021-AQG values (5 µg/m$^3$) [31]. The area considered was the former district of Cremona (northern Italy), and the period was the decade of 2010–2019. Although the study considered the decade of 2010–2019, we extrapolated data relating to the four-year period of interest in our study, i.e., 2016–2019. Therefore, in the period of 2016–2019, the percentages of deaths attributable to PM$_{2.5}$ exposure were 13.8% from natural causes, 17.6% from cardiovascular diseases, 15.1% from respiratory diseases, and 16.4% from lung cancer [31]. In Pisa, the percentages of deaths attributable to PM$_{2.5}$ exposure equal to 6.1% from natural causes, 7.1% from cardiovascular disease, 6.6% from respiratory diseases, and 8.4% from lung cancer were observed. Another study, conducted at global level, used the 2021-AQG value as counterfactual and estimated the deaths from natural causes attributable to NO$_2$ exposure between 2015 and 2019 in 13,169 worldwide urban areas [32]. The study found that 81% of people worldwide lived in cities with NO$_2$ levels exceeding the 2021-AQG values, in line with what was observed in Pisa municipality. Globally, the percentage of deaths from natural causes attributable to NO$_2$ exposure was 2.7% in 2019 [32], lower than Pisa's percentage (4.9%).

The methodology used to calculate the deaths attributable to air pollution has some limitations related to the assumptions made on the factors contained in the formula. First, since specific relative risks of the study area calculated with ad hoc studies are not available, the risk functions indicated by the WHO are used, assuming that these risks are representative of those present in the study area. On the other hand, the risk functions, being the result of reliable meta-analytic estimates, allow the attributable deaths to be accurately calculated even in areas where ad hoc studies to estimate the risks associated with air pollution have not been carried out. Furthermore, another assumption is that the regional rate is representative of the mortality risk of the area in the absence of exposure. This rate is reliable because it is based on a large population with characteristics like those of the study area. As reported in Stafoggia et al. (2020) [34], to assess the reliability of the method, the mean annual exposure calculated through the population-weighted exposure is compared with the mean annual concentrations provided by all the measurement points of the monitoring stations. Generally, the population-weighted exposure represents a good estimate of population exposure, especially in small-scale areas [34]. In fact, for particulate matter, the model is very high performing, providing an excellent goodness of fit of the estimated values compared to the measurements of the stations. For NO$_2$, however, a greater variability in the measured concentrations is observed compared to the model estimates due to its strong dependence on vehicular traffic [34]. Comparing the mean values of the mean concentrations in the period of 2016–2019 of the Pisa area with the population-weighted exposure, non-significant deviations are observed, confirming the validity of the use of population-weighted exposure in the study area.

Despite the limitations previously reported, it should be highlighted that it is the first time the model by Stafoggia et al. (2020) [34] is applied to NO$_2$. A further advantage lies in collecting uniform data at a national level; this allows studies to be carried out throughout the Italian territory with results that can be compared among different areas (regarding the population's exposure to air pollution). One of the strengths of our study is the use of the 2021-AQG values, unlike most studies that use the 2005-AQG values; furthermore, by counting the resident population for each cell into which the municipal territory of Pisa is divided and not using the census sections, more detailed information regarding both the population-weighted exposures and the attributable deaths are obtained.

Considering that 80% of the world's population lives in areas where even the highest limits set by previous guidelines have not been respected, it is now obvious that the work to be carried out by politicians, decision-makers, and citizens for the protection of health is considerable and fundamental, and policies aimed at reducing concentrations must be pursued. The benefits are clear: Reducing pollution levels will result in huge improvements in the health of people of all ages, who will breathe cleaner air.

As reported in the introduction, the IEHIA is a methodological approach that is particularly useful in providing an answer to the need for knowledge of civil society and

stakeholders, as it is a tool supporting environmental–health decision-making policies and planning since it helps in identifying the most impacted sub-areas that need mitigation actions. In fact, the secondary objective of this type of study is to provide indications to reduce the number of deaths among residents by promoting actions that reduce air pollution. In particular, municipalities can thus identify the most critical areas where action can be taken. Considering that vehicles powered by fossil fuels are among the major pollutants, municipalities could reduce air pollution by promoting the use of electric vehicles and the adoption of low-emission public transport. Municipalities can consider purifying the air by increasing the number of parks and green spaces within the city to absorb some of the pollutants. Promoting the construction of new buildings that respect the principles of smart building and that use sustainable construction methods is another strategy that helps improve air quality. Municipalities can resort to intelligent urban planning; in this case, technologies can be used to support urban planning that is more attentive to the needs of citizens and the environment. Promoting the grouping of buildings, services, and infrastructure, and thus reducing the need for vehicular travel, is a strategy that is becoming increasingly popular. This approach is the basis of the "15 min city", an urban model that hypothesizes the possibility for citizens to reach most places of interest within a quarter of an hour. Encouraging the adoption of similar measures as much as possible can improve the well-being of citizens. Even the "Pedibus" initiative, which aims to accompany children on foot to school under the supervision of an adult, starting in October 2024 in some schools in Pisa, goes in the direction of more sustainable mobility. At the local policy level, actions that could allow for a reduction in pollution and a consequent improvement in the health of residents are planning both the expansion of green spaces, which are a natural barrier to the transport of dust, and the cleaning of streets in urban areas characterized by a high population density and low rainfall. Obviously, the monitoring campaigns of airborne dust must continue to allow the planning of measures aimed at reducing the traffic of highly polluting vehicles.

This study tries to fill the gap regarding the need for the development of powerful tools to support priority-setting and guide policymakers in their choice of environmental policies. Considering this context, this study produced important information for policymakers to prioritize actions to investigate social health inequalities, such as the quantification of the number of deaths attributable to a reduction in $PM_{2.5}$, $PM_{10}$, and $NO_2$, and the spatial distribution of the health and equity impacts of reducing these three pollutants. Policymakers could use this approach to study different scenarios before and after an action they decide to implement to evaluate the benefits of applicable policies and identify the best strategies to pursue.

## 5. Conclusions

This study, carried out on the territory of Pisa municipality, has provided an IEHIA tool by identifying the most polluted areas that require mitigation actions and offering the municipal administration a support tool for both environmental and health policies and territorial planning (also in municipal operational plans). This assessment has provided an estimate of the exposure of residents to air pollution and the number of deaths attributable to the difference between air pollution in Pisa and the latest guideline values recommended by the WHO. According to the study, the distributions by cell of the deltas of the air pollutants considered highlighted that the areas of the urban center and some suburban areas are those with the highest values of air pollutants. Furthermore, the results for the entire territory show levels of air pollution, in terms of population-weighted exposure, that are significantly higher than the WHO reference levels, with a percentage of attributable deaths from natural causes of approximately 6% of the total mortality. This percentage is, however, lower than that of other Italian areas, such as the Po valley.

We would like to point out that it is necessary to interpret these results with due caution since these are estimates obtained by applying an analysis methodology that,

although consolidated, does not consider, as it is constituted, all the factors that can impact the health of citizens.

As specified in the previous paragraphs, one of the main objectives to be pursued is a reduction in emissions of pollutants into the atmosphere that represent an avoidable risk to health, cause a great burden for society in terms of death and other health outcomes, and, consequently, have an enormous social and economic cost.

In the awareness of how actions to reduce air pollution require cooperation between various sectors and stakeholders, an analysis of the health effects exerted by air pollution appears essential to increase awareness of the population and ensure that health protection is a determining aspect in the political debate and administrative choices. What is needed, therefore, is a shift in perspective from using fixed-limit values alone to a concept of combining fixed-limit values with a continuous reduction in average exposure.

Indeed, given today's budgetary constraints, it can be quite challenging for policy-makers to select an initiative. Programs that reduce emissions of pollutants provide huge benefits for both air quality and health that increase over time. The estimated health benefits of improving air quality far outweigh the costs of implementing actions to achieve air quality improvement. This shows the need for tools to support priority-setting and to guide policymakers in their choice of environmental initiatives that would maximize health gains and reduce social inequalities in health.

**Author Contributions:** Conceptualization, E.B. and F.M.; data curation, E.B. and F.M.; formal analysis, F.M.; funding acquisition, M.R. and F.M.; methodology, F.M.; project administration, F.M.; software, E.B. and F.M.; supervision, F.M.; validation, E.B. and F.M.; visualization, E.B.; writing—original draft, E.B. and F.M.; writing—review and editing, E.B., M.R. and F.M. All authors have read and agreed to the published version of the manuscript.

**Funding:** This work was supported by the Municipality of Pisa Town through the Convention with the Institute of Clinical Physiology of the National Research Council of Italy (IFC-CNR) (project DSB.AD008.543; project area DSB.AD008; GAE P002260. Funder: Municipality of Pisa).

**Institutional Review Board Statement:** Not applicable.

**Informed Consent Statement:** Not applicable.

**Data Availability Statement:** 3rd Party Data. Restrictions apply to the availability of these data. Data was obtained from DEP Lazio and Municipality of Pisa and are available from the author Minichilli Fabrizio with the permission of DEP Lazio and Municipality of Pisa.

**Acknowledgments:** The authors would like to thank Massimo Stafoggia, Lazio Region Health Service, Rome, for providing $PM_{2.5}$, $PM_{10}$, and $NO_2$ concentration maps; and Cristina Imiotti, Daniela Banti, and Anna Paola Pala, Institute of Clinical Physiology of NRC Pisa, for the support of technical and secretarial activities.

**Conflicts of Interest:** The authors declare no conflicts of interest.

## List of Acronyms

| | |
|---|---|
| IEHIA | Integrated Environmental and Health Impact Assessment |
| $NO_2$ | Nitrogen dioxide |
| PM | Particulate matter |
| $PM_{2.5}$ | Particulate matter with a diameter of <2.5 microns |
| $PM_{10}$ | Particulate matter with a diameter of <10 microns |
| WHO | World Health Organization |
| 2021-AQG | WHO Air Quality Guidelines, 2021 |
| 2005-AQG | WHO Air Quality Guidelines, 2005 |

## Appendix A

(i)    Mortality and percentage of mortality attributable to the delta of the target pollutants for the selected causes of death, using as counterfactual the 2005-AQG values.

**Table A1.** Mortality and percentage of mortality attributable to the delta of the target pollutants for the selected causes of death (using as counterfactuals the 2005-AQG values).

| Cause of Death | Pollutant * | Annual Average of Attributable Deaths (CI95%) ** | AD% (CI95%) |
|---|---|---|---|
| Natural causes | $PM_{2.5}$ | 35.9 (27.2–40.2) | 3.4 (2.6–3.9) |
| | $PM_{10}$ | 14.8 (11.1–22.0) | 1.4 (1.1–2.1) |
| | $NO_2$ | −51.2(−21.3–(−83.4)) | −4.9 (−2.0–(−8.0)) |
| Cardiovascular diseases | $PM_{2.5}$ | 14.5 (12.0–18.1) | 4.1 (3.4–5.1) |
| Ischemic diseases | $PM_{10}$ | 1.6 (0.3–2.7) | 1.5 (0.3–2.6) |
| Respiratory diseases | $PM_{2.5}$ | 3.3 (1.0–5.7) | 3.8 (1.2–6.6) |
| | $PM_{10}$ | 3.2 (1.6–4.9) | 3.7 (1.9–5.6) |
| | $NO_2$ | −2.5 (−0.8–(−4.4)) | −2.9 (−1.0–(−5.0)) |
| Lung cancer | $PM_{2.5}$ | 2.9 (1.7–3.7) | 4.9 (2.9–6.4) |
| | $PM_{10}$ | 1.6 (0.8–2.5) | 2.7 (1.4–4.3) |

Legend—*: The guideline values indicated by the World Health Organization in 2005 are 10 $\mu g/m^3$, 20 $\mu g/m^3$, and 40 $\mu g/m^3$ for $PM_{2.5}$, $PM_{10}$, and $NO_2$, respectively; CI95%: confidence interval at 95% of probability; **: to calculate the lower and upper limits of this CI95%, the confidence interval limits of the concentration–response functions were used; AD%: percentage of deaths attributable to exposure to the pollutants under study compared to the total average number of deaths in the period of 2016–2019 in the municipality of Pisa.

(ii) Mortality distribution per cell for each cause and for each pollutant considered

Mortality from natural causes—In Figure A1, it can be observed that the cell distribution of mortality from natural causes attributable to population exposure to the pollutants under study is mainly concentrated in urban areas for PM and in urban and suburban areas for $NO_2$. This distribution, which for all pollutants has a minimum of 0 attributable deaths per cell, for $PM_{2.5}$ has an average of 0.3, with a maximum of 5.8; for $PM_{10}$ an average of 0.1, with a maximum of 2.9; and for $NO_2$ an average of 0.2, with a maximum of 5.8.

Mortality from cardiovascular diseases—In Figure A2, it can be observed that the distribution by cell of mortality from cardiovascular diseases attributable to the population's exposure to $PM_{2.5}$ is mainly concentrated in urban areas and shows an average of 0.1 attributable deaths per cell, with a minimum and maximum of 0 and 2.3, respectively.

Mortality from ischemic diseases—In Figure A3, it can be observed that the distribution by cell of mortality from ischemic diseases attributable to the population's exposure to $PM_{10}$ is mainly concentrated in urban areas and shows an average of 0 attributable deaths per cell, with a minimum and maximum of 0 and 0.3, respectively.

Mortality from respiratory diseases—In Figure A4, it can be observed that the cell distribution of mortality from respiratory diseases attributable to the exposure of the population to the pollutants under study is mainly concentrated in urban areas for PM and in urban and suburban areas for $NO_2$. This distribution, which for all pollutants has a mean and a minimum equal to 0 attributable deaths per cell, for $PM_{2.5}$ presents a maximum of 0.5, for $PM_{10}$ a maximum of 0.6, and for $NO_2$ a maximum of 0.3.

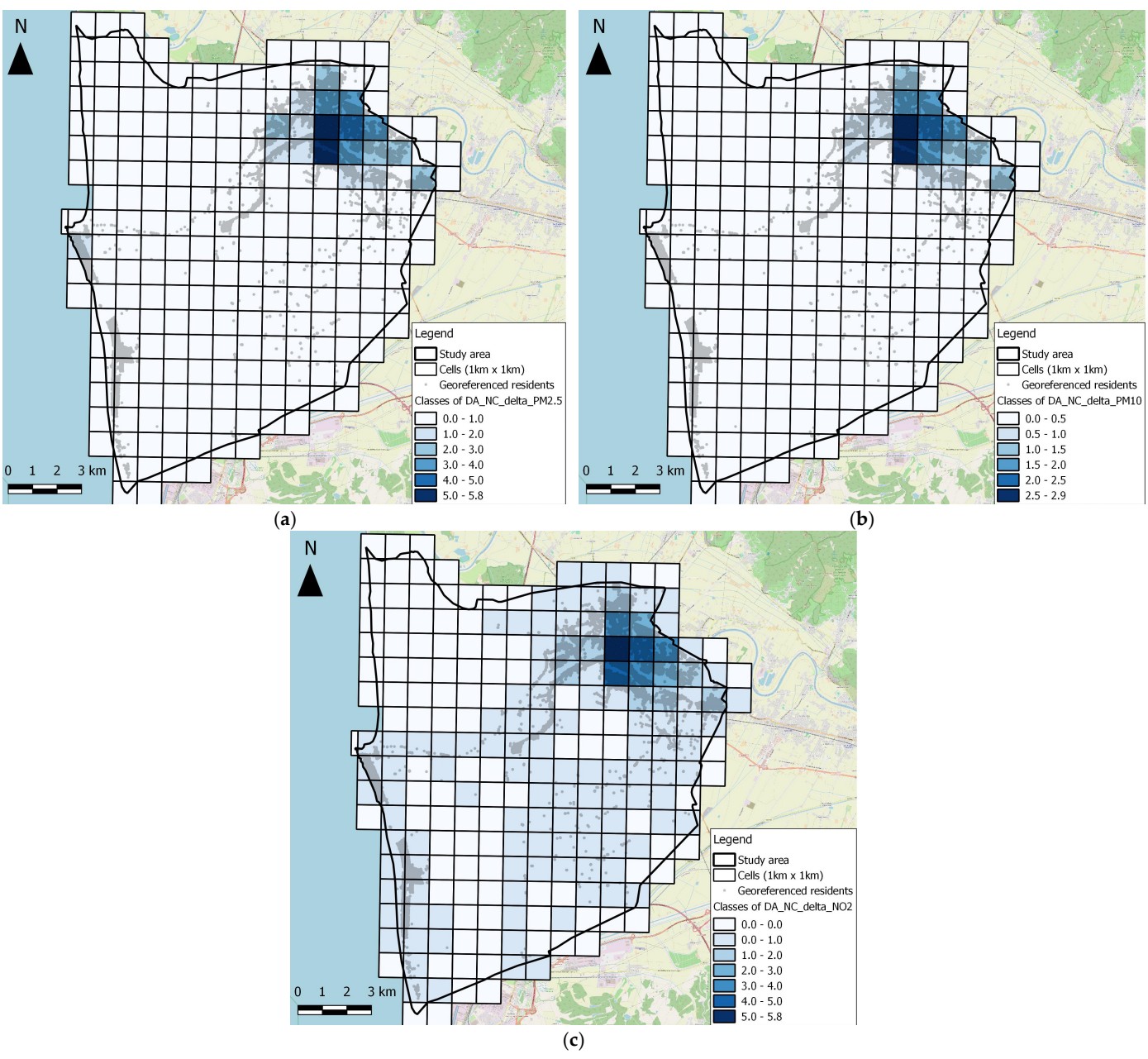

**Figure A1.** Distribution of mortality from natural causes attributable to exposure to PM$_{2.5}$ (**a**), PM$_{10}$ (**b**), and NO$_2$ (**c**), using as counterfactual the values defined by the World Health Organization in 2021 (5 µg/m$^3$ for PM$_{2.5}$, 15 µg/m$^3$ for PM$_{10}$, and 10 µg/m$^3$ for NO$_2$).

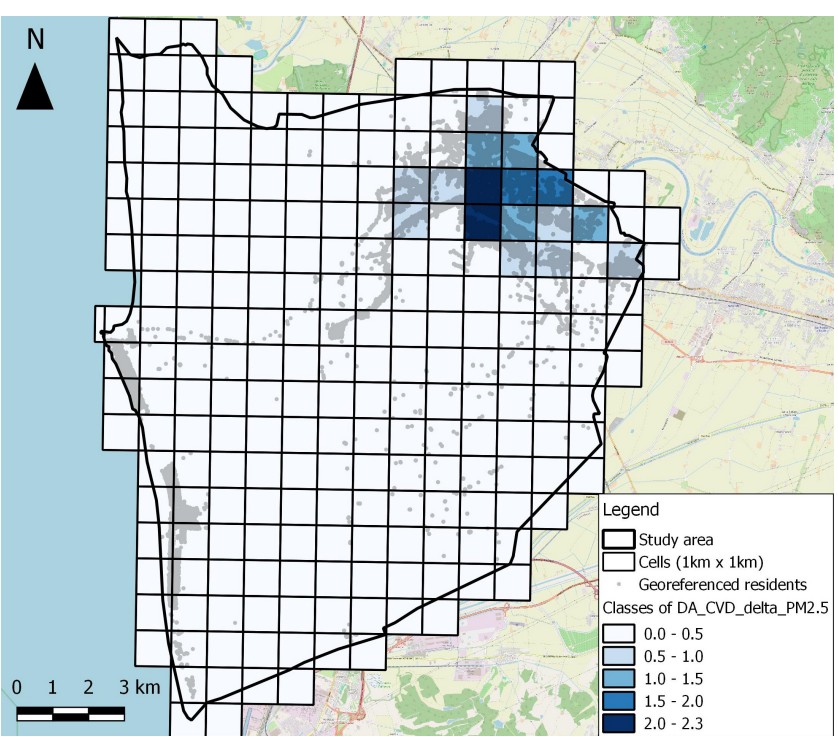

**Figure A2.** Distribution of mortality from cardiovascular diseases attributable to exposure to $PM_{2.5}$, using as counterfactual the value defined by the World Health Organization in 2021 of 5 µg/m³.

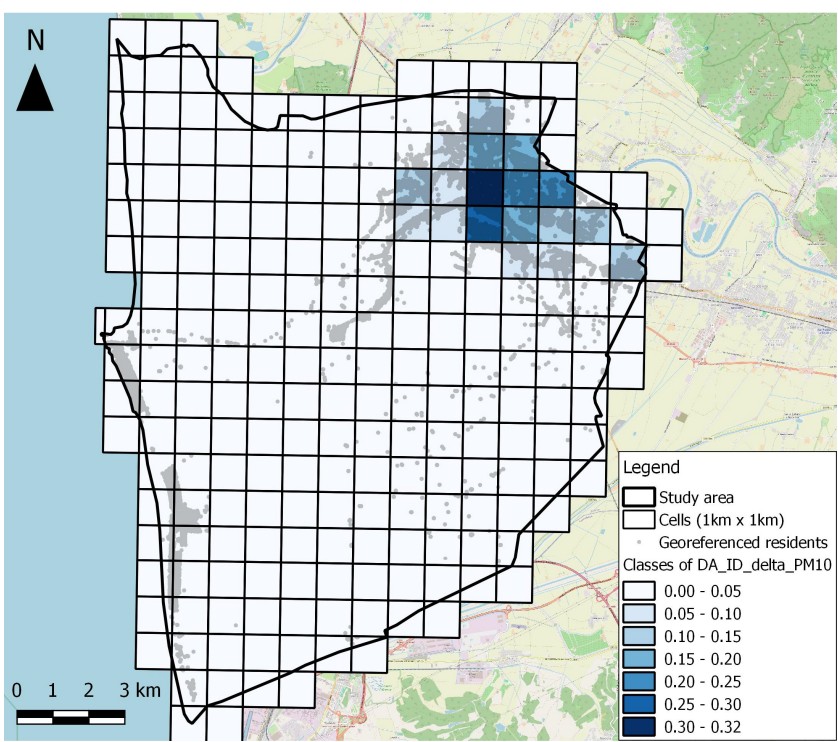

**Figure A3.** Distribution of mortality from ischemic diseases attributable to $PM_{10}$ exposure, using as counterfactual the value defined by the World Health Organization in 2021 of 15 µg/m³.

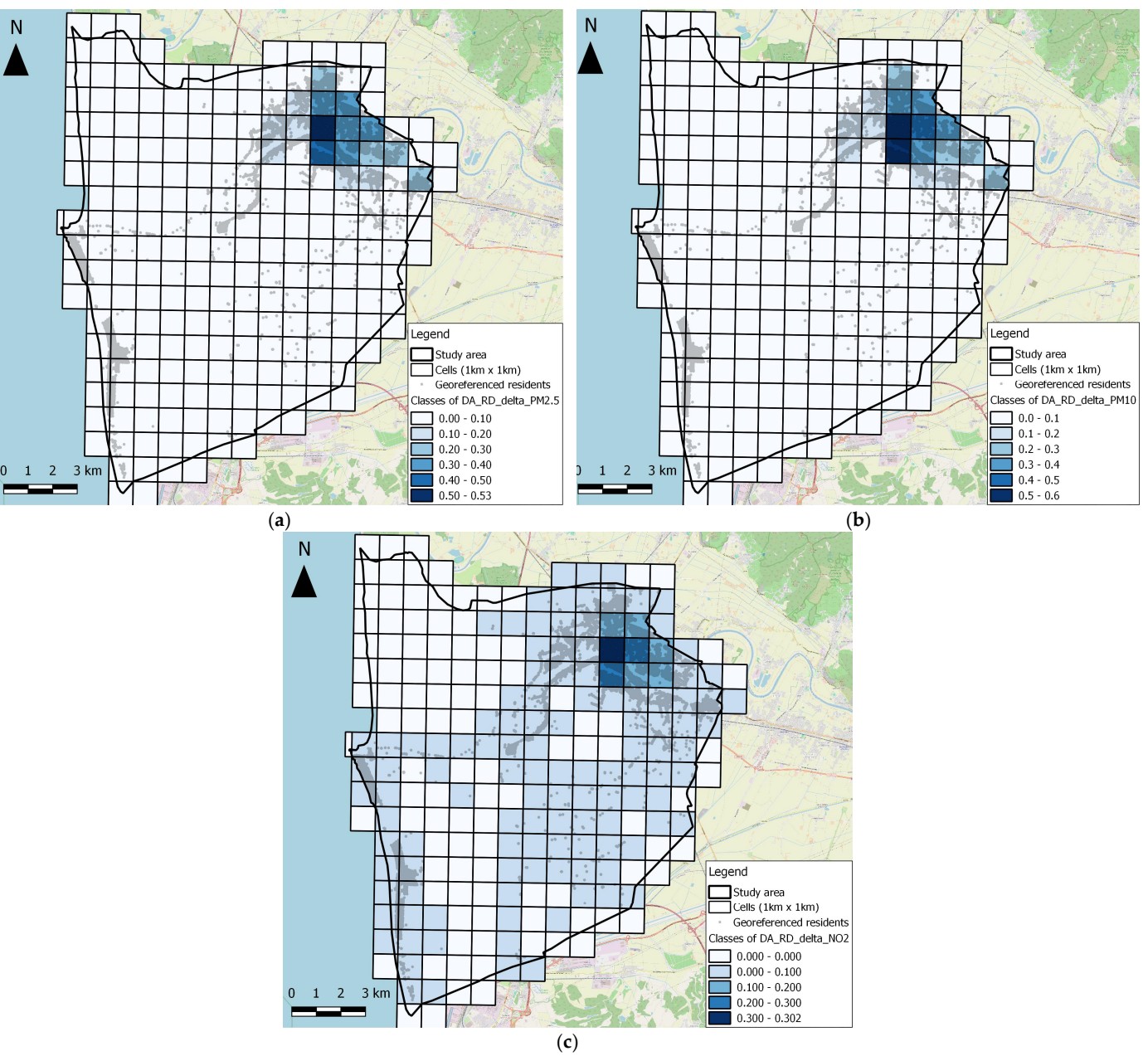

**Figure A4.** Distribution of mortality for respiratory diseases attributable to exposure to $PM_{2.5}$ (**a**), to $PM_{10}$ (**b**), and to $NO_2$ (**c**), using as counterfactual the values defined by the World Health Organization in 2021 (5 µg/m$^3$ for $PM_{2.5}$, 15 µg/m$^3$ for $PM_{10}$, and 10 µg/m$^3$ for $NO_2$).

Mortality form lung cancer—In Figure A5, it can be observed that the cell distribution from lung cancer mortality attributable to the population's exposure to the pollutants under study is mainly concentrated in urban areas. This distribution, which for both pollutants shows a mean and a minimum equal to 0 attributable deaths per cell, for $PM_{2.5}$ presents a maximum of 0.5 and for $PM_{10}$ a maximum of 0.3.

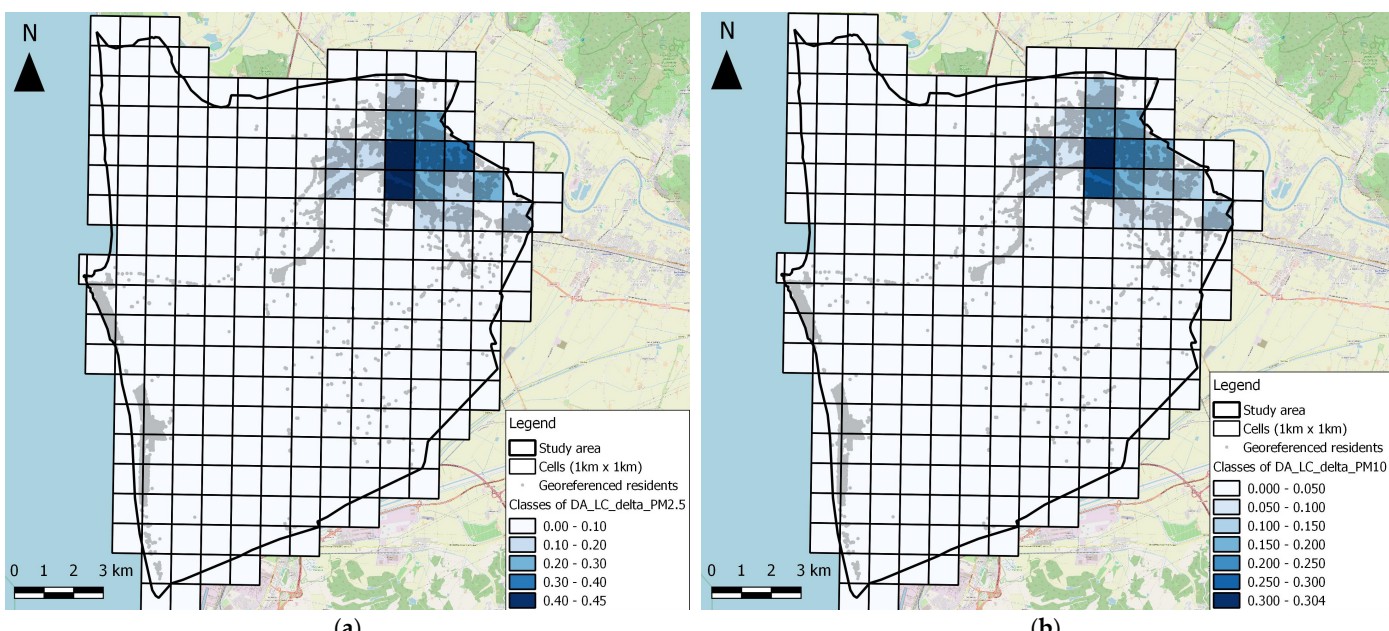

**Figure A5.** Distribution of lung cancer mortality attributable to exposure to PM$_{2.5}$ (**a**) and PM$_{10}$ (**b**) using as counterfactual the values defined by the World Health Organization in 2021 (5 μg/m$^3$ for PM$_{2.5}$ and 15 μg/m$^3$ for PM$_{10}$).

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
