# Peer review of "Assessment of Mortality Attributable to Air Pollution in the Urban Area of Pisa (Central Italy) Characterized by Multi-Source Exposures"

_atmosphere, doi:10.3390/atmos15111311_

Round 1
Reviewer 1 Report
Comments and Suggestions for Authors
Review of Manuscript: Assessment of Mortality Attributable to Air Pollution in a Urban Area Characterized by Multi-Source Exposures
Summary
The manuscript addresses an important issue: the estimation of mortality attributable to air pollution in Pisa, Italy, utilizing an Integrated Environmental and Health Impact Assessment (IEHIA) tool. The study focuses on pollutants such as PM2.5, PM10, and NO2, analyzing their effects on human health by calculating Population Weighted Exposure (PWE) and Attributable Deaths (ADs). The results underline the necessity of mitigation actions in urban areas with significant environmental pressures, contributing to better policy-making in the field of environmental health.
General Comments
This manuscript tackles a highly relevant subject with significant implications for public health policies. The methodology is robust, and the use of updated WHO guidelines adds credibility. However, there are certain areas that need refinement to strengthen the scientific soundness and the presentation of results.
1. Clarity and Focus: The manuscript's objectives are clear, but at certain points, the writing lacks clarity. The narrative could benefit from more concise explanations, particularly in the methodology section.
2. Scientific Rigor: The methodology is scientifically sound, but there is a lack of discussion regarding the uncertainties or limitations in the model used (i.e., Stafoggia et al. 2020). Addressing these uncertainties could enhance the robustness of the findings.
3. Data Representation: The figures and tables are informative, but some of them, such as Figures 3–5, could benefit from clearer legends and more detailed captions to ensure they are understandable without excessive reference to the main text. Also, consider including confidence intervals or a discussion of statistical significance where applicable.
4. Novelty: While the study's findings are regionally important, the contribution to global literature would benefit from a stronger emphasis on how this study differs from or improves upon similar studies conducted in other cities or regions.
Specific Comments
1. Title: The title could be more specific by including the geographical region (Pisa, Italy) and specifying that this is an urban case study.
2. Introduction (Lines 26–85): The introduction provides a good overview of the issue, but it would benefit from a deeper analysis of existing literature on air pollution in Italy or similar urban areas. This would help to position the study more clearly within the current research landscape.
3. Methodology (Lines 86–239):
The methodology is well-structured, but a more detailed explanation of the choice of pollutants (PM2.5, PM10, NO2) and their relevance to Pisa would be helpful.
The formulae (1) and (2) are well explained, but the manuscript would benefit from more discussion of any assumptions made in these calculations. Additionally, could the authors clarify if any validation of the PWE estimates was performed using observed concentration data?
4. Results (Lines 266–319):
The results are comprehensive, but some sections, such as the description of PM10 results, are brief and could use more in-depth discussion.
Furthermore, it would be useful to see a comparison of Pisa’s results with other regions in Italy or Europe.
5. Discussion (Lines 328–434):
The discussion would benefit from a more thorough examination of the potential policy implications of the findings. While the study does touch on this, a more detailed section on how municipalities can use these results to mitigate air pollution could enhance the practical application of the research.
It would be useful to include a more detailed analysis of how the ADs in Pisa compare to other urban centers, both regionally and globally, to highlight the broader significance of the findings.
6. Conclusion (Lines 435–475):
The conclusion is well-written but could benefit from stronger recommendations for future research or policies, including any gaps identified in the current study.
Overall Recommendation
I recommend Reconsider after Major Revisions. The study is valuable and addresses a critical issue, but it would benefit from further discussion on the uncertainties, clearer data presentation, and a more detailed comparison with existing literature.
Author Response
Comments and Suggestions for Authors
Review of Manuscript: Assessment of Mortality Attributable to Air Pollution in a Urban Area Characterized by Multi-Source Exposures
Summary
The manuscript addresses an important issue: the estimation of mortality attributable to air pollution in Pisa, Italy, utilizing an Integrated Environmental and Health Impact Assessment (IEHIA) tool. The study focuses on pollutants such as PM2.5, PM10, and NO2, analyzing their effects on human health by calculating Population Weighted Exposure (PWE) and Attributable Deaths (ADs). The results underline the necessity of mitigation actions in urban areas with significant environmental pressures, contributing to better policy-making in the field of environmental health.
The authors would like to thank the reviewer for his/her valuable comments and suggestions. Overall, the text has been deeply revised and modified; entire blocks of text have been changed, especially in the methodology and the discussion sections, also based on the comments of the other two reviewers. The maps have all been changed to make the distribution of the population within the territory clearer and thus make the graphic representation of each map more comprehensible to the reader. The title and the bibliography have been modified, the objectives, limits and advantages and importance of the study at not only a regional but also a global level have been made clearer. To make comparisons with the available scientific literature possible, analyses using as counterfactuals also the air quality values indicated by the World Health Organization in 2005 have been performed.
General Comments
This manuscript tackles a highly relevant subject with significant implications for public health policies. The methodology is robust, and the use of updated WHO guidelines adds credibility. However, there are certain areas that need refinement to strengthen the scientific soundness and the presentation of results.
- Clarity and Focus: The manuscript's objectives are clear, but at certain points, the writing lacks clarity. The narrative could benefit from more concise explanations, particularly in the methodology section.
We modified the methodology section to attempt the reviewer suggestion (lines 125-148).
- Scientific Rigor: The methodology is scientifically sound, but there is a lack of discussion regarding the uncertainties or limitations in the model used (i.e., Stafoggia et al. 2020). Addressing these uncertainties could enhance the robustness of the findings.
We added a paragraph in the discussion section (lines 484-505).
- Data Representation: The figures and tables are informative, but some of them, such as Figures 3–5, could benefit from clearer legends and more detailed captions to ensure they are understandable without excessive reference to the main text.
We thank the reviewer for this observation. We modified Figures 3-5 and rewrote the captions in a clearer way.
Also, consider including confidence intervals or a discussion of statistical significance where applicable.
We have included reference to confidence intervals in the discussion where possible (lines 407-416).
- Novelty: While the study's findings are regionally important, the contribution to global literature would benefit from a stronger emphasis on how this study differs from or improves upon similar studies conducted in other cities or regions.
We modified the discussion and conclusion section in order to attempt the reviewer suggestion (lines 507-514).
Specific Comments
- Title: The title could be more specific by including the geographical region (Pisa, Italy) and specifying that this is an urban case study.
We modified the title in “Assessment of Mortality Attributable to Air Pollution in the Urban Area of Pisa (central Italy) Characterized by Multi-Source Exposure”.
- Introduction (Lines 26–85): The introduction provides a good overview of the issue, but it would benefit from a deeper analysis of existing literature on air pollution in Italy or similar urban areas. This would help to position the study more clearly within the current research landscape.
We understand the reviewer's point of view; however, the aim of this study is to provide a useful tool for decision makers; a tool that provides the number of deaths that would be avoided if we could reach the pollutants concentration values indicated by the WHO. This is directly linked to the average concentrations of pollutants in Pisa but instead is directly linked to the population-weighted exposure, which is widely described and reported in the text. We therefore believe that it is not necessary to report in the introduction a description of the average concentrations of pollutants in the Pisa’s atmosphere. Anyway, we reported in the method section the comparison between the average pollution levels in Pisa and the regional situation (lines 198-209).
- Methodology (Lines 86–239):
The methodology is well-structured, but a more detailed explanation of the choice of pollutants (PM2.5, PM10, NO2) and their relevance to Pisa would be helpful.
We modified the text in the methods section (lines 136-146).
The formulae (1) and (2) are well explained, but the manuscript would benefit from more discussion of any assumptions made in these calculations. Additionally, could the authors clarify if any validation of the PWE estimates was performed using observed concentration data?
We modified the discussion section in order to attempt the reviewer suggestion (ex 484-505).
- Results (Lines 266–319):
The results are comprehensive, but some sections, such as the description of PM10 results, are brief and could use more in-depth discussion.
We have added comments for all pollutant results both at the exposure data level and for health impacts.
Furthermore, it would be useful to see a comparison of Pisa’s results with other regions in Italy or Europe.
We reported in the method section the comparison between the average pollution levels in Pisa and the regional situation (lines 198-209). In discussion section we added comments regarding suggested comparisons (lines 416-519). Furthermore, to make comparisons between Pisa and other countries, the analyses using as counterfactual the 2005-AQG values have been carried out. Therefore, comments relating to comparisons between the situation of Pisa and other realities that were not present before are reported in the discussion section (lines 433-483).
- Discussion (Lines 328–434):
The discussion would benefit from a more thorough examination of the potential policy implications of the findings. While the study does touch on this, a more detailed section on how municipalities can use these results to mitigate air pollution could enhance the practical application of the research.
We added a paragraph in the discussion section (lines 532-534).
It would be useful to include a more detailed analysis of how the ADs in Pisa compare to other urban centers, both regionally and globally, to highlight the broader significance of the findings.
We revised the discussion section (lines 433-483).
- Conclusion (Lines 435–475):
The conclusion is well-written but could benefit from stronger recommendations for future research or policies, including any gaps identified in the current study.
We added a paragraph in the conclusion section (lines 591-594).
Overall Recommendation
I recommend Reconsider after Major Revisions. The study is valuable and addresses a critical issue, but it would benefit from further discussion on the uncertainties, clearer data presentation, and a more detailed comparison with existing literature.
We trust that the changes made have improved the article as indicated by the reviewer.

Reviewer 2 Report
Comments and Suggestions for Authors
This paper presents a novel Integrated Environmental and Health Impacts Assessment (IEHIA) tool applied to Pisa, Italy. This tool estimates the impacts of air pollution (PM2.5, PM10, and NO2) on human health by calculating Population Weighted Exposure (PWE) and Attributable Deaths (ADs). The study reveals significantly higher pollution levels than WHO guidelines, attributing approximately 6% of total mortality in Pisa to air pollution. The novelty lies in its detailed spatial analysis and the use of updated WHO guidelines, providing crucial insights for environmental and health policy-making aimed at reducing pollution and associated mortality.
To improve the paper the authors need to address the following information:
- It is not clear why this research is new. There is no a section in the introduction where it is explained other previous works similar to this where they use this kind of tools to determine the effect of air pollution in health. They have to explain why their paper is innovative.
- Indoor air pollution would affect as much as outdoor pollution. Please comment and consider that in the paper.
- In line 211 the authors comment: "The calculation of the ADs was carried out by applying a methodology extracted 211 from some international and national publications [29–32] using the following formula:". Please clarify this, which are these international and national publications and why are relevant for this work?
- The paper is full of acronyms, so it is really easy to get lost. Please add a list of acronyms.
Comments on the Quality of English LanguageAlthough English is good enough to be understood the authors should revise it. Below some examples of places where English can be improved:
* Line 13 “considering the difference among the PWE and the latest air quality guidelines” “among” should be “between”.
* Line 21: “in critical area with the consequent reduction in avoidable mortality.” “area” should be plural.
* Line 27 "one of the greater risks to human health", “greater” should be “greatest”.
* Line 31: “because of advances in air pollution measurement and exposure assessment with large monitoring datasets.” The sentence does not sound OK, please rewrite it.
* Line 37: “even if long-term exposures have much more significant impacts on public health since even exposures at very low levels can cause negative effects” Rewrite as follows: "“although long-term exposures have much more significant impacts on public health, as exposures at very low levels can cause negative effects”
* Line 45: “the global role played by the WHO Air Quality Guidelines (AQG)” It sound very informal. Change by "“the global importance of the WHO Air Quality Guidelines (AQG)”
There are much many issues like this. So, please make a whole revision of the English in the document.
Author Response
Comments and Suggestions for Authors
This paper presents a novel Integrated Environmental and Health Impacts Assessment (IEHIA) tool applied to Pisa, Italy. This tool estimates the impacts of air pollution (PM2.5, PM10, and NO2) on human health by calculating Population Weighted Exposure (PWE) and Attributable Deaths (ADs). The study reveals significantly higher pollution levels than WHO guidelines, attributing approximately 6% of total mortality in Pisa to air pollution. The novelty lies in its detailed spatial analysis and the use of updated WHO guidelines, providing crucial insights for environmental and health policy-making aimed at reducing pollution and associated mortality.
The authors would like to thank the reviewer for his/her valuable comments and suggestions. Overall, the text has been deeply revised and modified; entire blocks of text have been changed, especially in the methodology and the discussion sections, also based on the comments of the other two reviewers. The maps have all been changed to make the distribution of the population within the territory clearer and thus make the graphic representation of each map more comprehensible to the reader. The title and the bibliography have been modified, the objectives, limits and advantages and importance of the study at not only a regional but also a global level have been made clearer. To make comparisons with the available scientific literature possible, analyses using as counterfactuals also the air quality values indicated by the World Health Organization in 2005 have been performed.
We trust that the changes made have improved the article as indicated by the reviewer.
To improve the paper the authors need to address the following information:
- It is not clear why this research is new. There is no a section in the introduction where it is explained other previous works similar to this where they use this kind of tools to determine the effect of air pollution in health. They have to explain why their paper is innovative.
We added a paragraph in the introduction in order to attempt the reviewer suggestion (lines 86-91).
- Indoor air pollution would affect as much as outdoor pollution. Please comment and consider that in the paper.
We agree with the reviewer but unfortunately the methodology does not include the measurement of indoor pollution. The concentration-response functions, on which the calculation of attributable deaths is based, are the results of meta-analyses of studies that do not consider indoor pollution.
- In line 211 the authors comment: "The calculation of the ADs was carried out by applying a methodology extracted 211 from some international and national publications [29–32] using the following formula:". Please clarify this, which are these international and national publications and why are relevant for this work?
We simply applied a formula that has also been used in other studies, now validated and widely used at national and international level, as reported in the bibliography. We modified the text in the manuscript to make it clearer.
- The paper is full of acronyms, so it is really easy to get lost. Please add a list of acronyms.
We agree with the reviewer. To make the text easier to read we decided to keep the chemical acronyms/symbols only for the pollutants, for the World Health Organization, and for the Air Quality Guidelines and to make the rest explicit. However, we added the list of acronyms used at the end of the text.
Comments on the Quality of English Language
Although English is good enough to be understood the authors should revise it. Below some examples of places where English can be improved:
* Line 13 “considering the difference among the PWE and the latest air quality guidelines” “among” should be “between”.
* Line 21: “in critical area with the consequent reduction in avoidable mortality.” “area” should be plural.
* Line 27 "one of the greater risks to human health", “greater” should be “greatest”.
* Line 31: “because of advances in air pollution measurement and exposure assessment with large monitoring datasets.” The sentence does not sound OK, please rewrite it.
* Line 37: “even if long-term exposures have much more significant impacts on public health since even exposures at very low levels can cause negative effects” Rewrite as follows: "“although long-term exposures have much more significant impacts on public health, as exposures at very low levels can cause negative effects”
* Line 45: “the global role played by the WHO Air Quality Guidelines (AQG)” It sound very informal. Change by "“the global importance of the WHO Air Quality Guidelines (AQG)”
There are much many issues like this. So, please make a whole revision of the English in the document.
We completely revised the English in throughout the text.

Reviewer 3 Report
Comments and Suggestions for Authors
Dear Authors!
Thank you for submitting your manuscript “Assessment of Mortality Attributable to Air Pollution in an Urban Area Characterized by Multi-Source Exposures” to the Journal of “Atmosphere”.
Overall, the work is done at a fairly high level and can be published in a journal after revision.
- Line 14. Bring units of measurement to a single dimension (we are talking about tenths and hundredths)
- Lines 88-93. As an explanation for me. Are there industrial facilities in the city of Pisa or its environs? And where are they located? In the north-eastern part of the map?
- Figure 1. It would be useful to add a general map of the study region to the figure indicating the scale (applies to all maps) and the direction to the north (0 degrees), which also applies to all maps.
- You periodically mention O3, SO2, CO as parameters under consideration, or as pollutants that affect health. However, you do not use it for a full-fledged analysis. I think that since you do not study the relationship between O3, SO2, CO and the incidence rate, then the information about O3, SO2, CO should be removed. Or the article should be supplemented with an analysis of these components.
- Figures 3-5. It is necessary to use uniform designations of μg/m3 in the manuscript and figures to it. In addition, the figure should be supplemented by indicating the boundaries of the city and villages on it.
- As an addition, I would like to know if there is data on the recurrence of episodes of "extreme" pollution? (several times higher than average)
- I believe that the work will be improved if it analyzes meteorological parameters: wind conditions, humidity, precipitation, air temperature, number of calm situations. What do you think about this?
- I would like to clarify whether there is a temporal variability in pollution and morbidity in Pisa over the study period? - I would like to clarify whether there is data on the chemical composition of PM2.5 and PM10?
- I believe that it would be appropriate to conduct a comparative analysis with the level of PM air pollution in Pisa and other regions of Italy and the World. This would enhance the relevance of the research you are conducting.
For reference, I advise you to pay attention to the following works
Yousefian F., Faridi S., Azimi F., Aghaei M., Shamsipour M., Yaghmaeian K., Sadegh Hassanvand M. Temporal variations of ambient air pollutants and meteorological influences on their concentrations in Tehran during 2012–2017 // Sci. Report. 2020. V. 10, N 1. P. 1–11.
Shikhovtsev, M. Y., Obolkin, V. A., Khodzher, T. V., & Molozhnikova, Y. V. (2023). Variability of the Ground Concentration of Particulate Matter PM1–PM10 in the Air Basin of the Southern Baikal Region. Atmospheric and Oceanic Optics, 36(6), 655-662 Ma J. et al. NOx promotion of SO2 conversion to sulfate: An important mechanism for the occurrence of heavy haze during winter in Beijing //Environmental Pollution. - 2018. - Vol. 233. - Pp. 662-669. And a rural area under the intermittent influence of industrial facilities Your research is relevant and valuable certainly to the scientific community. Despite this, I must recommend that you improve some aspects of your article.
I understand that these comments may require additional effort on your part. However, addressing these issues will improve the quality and impact of your manuscript. I look forward to seeing an improved version of your manuscript in the future.
Author Response
Comments and Suggestions for Authors
Dear Authors!
Thank you for submitting your manuscript “Assessment of Mortality Attributable to Air Pollution in an Urban Area Characterized by Multi-Source Exposures” to the Journal of “Atmosphere”.
Overall, the work is done at a fairly high level and can be published in a journal after revision.
The authors would like to thank the reviewer for his/her valuable comments and suggestions. Overall, the text has been deeply revised and modified; entire blocks of text have been changed, especially in the methodology and the discussion sections, also based on the comments of the other two reviewers. The maps have all been changed to make the distribution of the population within the territory clearer and thus make the graphic representation of each map more comprehensible to the reader. The title and the bibliography have been modified, the objectives, limits and advantages and importance of the study at not only a regional but also a global level have been made clearer. To make comparisons with the available scientific literature possible, analyses using as counterfactuals also the air quality values indicated by the World Health Organization in 2005 have been performed.
We trust that the changes made have improved the article as indicated by the reviewer.
- Line 14. Bring units of measurement to a single dimension (we are talking about tenths and hundredths)
We modified the text of the abstract.
- Lines 88-93. As an explanation for me. Are there industrial facilities in the city of Pisa or its environs? And where are they located? In the north-eastern part of the map?
We specified this when we talked about Figure 2(b). Specifically, the industrial/commercial/artisanal activities are those colored in pink and blue in Figure 2 and are therefore mainly distributed outside of the historic center in the western and south-western part of the city of Pisa. To make this aspect clearer we modified the text (Lines 106-118).
- Figure 1. It would be useful to add a general map of the study region to the figure indicating the scale (applies to all maps) and the direction to the north (0 degrees), which also applies to all maps.
We modified all the figures in the manuscript.
- You periodically mention O3, SO2, CO as parameters under consideration, or as pollutants that affect health. However, you do not use it for a full-fledged analysis. I think that since you do not study the relationship between O3, SO2, CO and the incidence rate, then the information about O3, SO2, CO should be removed. Or the article should be supplemented with an analysis of these components.
We talk about O3, SO2 and CO only in Table 1 when we report the latest air quality guidelines suggested by the World Health Organization compared to the Italian legislation values. Since these parametera are not actually taken into account in the study, we thought it was appropriate to eliminate them from Table 1.
- Figures 3-5. It is necessary to use uniform designations of μg/m3 in the manuscript and figures to it. In addition, the figure should be supplemented by indicating the boundaries of the city and villages on it.
The software we used do not allow us to insert quotes, so in the figures we are forced to write “μg/m3” instead of the correct form “μg/m3”. The boundaries of the city are indicated as reported in the legend whithin the figure.
- As an addition, I would like to know if there is data on the recurrence of episodes of "extreme" pollution? (several times higher than average)
The data are available in the archives of the Regional Agency for Environmental Protection of Tuscany but are not necessary for our purpose which is to evaluate the average exposure of the population and the medium-long term effects as we have specified in the article.- I believe that the work will be improved if it analyzes meteorological parameters: wind conditions, humidity, precipitation, air temperature, number of calm situations. What do you think about this?
In the methodology used by Stafoggia to build the concentration map, meteorological data were considered as reported in the text: “This methodology integrates meteorological, land use and monitoring data obtained from all available sites provided by the Higher Institute for Environmental Research and Protection together with satellite data on the optical depth of aerosols.”
- I would like to clarify whether there is a temporal variability in pollution and morbidity in Pisa over the study period? - I would like to clarify whether there is data on the chemical composition of PM2.5 and PM10?
As written in the text there is a slight decrease in concentrations from 2016 to 2019 although not significant. Same consideration for mortality rates (lines 419-421).
In Tuscany the regional agency for environmental protection does not carry out the analysis of the fraction of particulate matter. We are aware of only one study that analyses the fraction of PM2.5 only in 3 selected stations, two in Florence and one in Livorno. We have no indications on Pisa.
- I believe that it would be appropriate to conduct a comparative analysis with the level of PM air pollution in Pisa and other regions of Italy and the World. This would enhance the relevance of the research you are conducting.
We modified the results and the discussion section to attempt the reviewer suggestion.
For reference, I advise you to pay attention to the following works
Yousefian F., Faridi S., Azimi F., Aghaei M., Shamsipour M., Yaghmaeian K., Sadegh Hassanvand M. Temporal variations of ambient air pollutants and meteorological influences on their concentrations in Tehran during 2012–2017 // Sci. Report. 2020. V. 10, N 1. P. 1–11.
Shikhovtsev, M. Y., Obolkin, V. A., Khodzher, T. V., & Molozhnikova, Y. V. (2023). Variability of the Ground Concentration of Particulate Matter PM1–PM10 in the Air Basin of the Southern Baikal Region. Atmospheric and Oceanic Optics, 36(6), 655-662 Ma J. et al. NOx promotion of SO2 conversion to sulfate: An important mechanism for the occurrence of heavy haze during winter in Beijing //Environmental Pollution. - 2018. - Vol. 233. - Pp. 662-669. And a rural area under the intermittent influence of industrial facilities Your research is relevant and valuable certainly to the scientific community. Despite this, I must recommend that you improve some aspects of your article.
I understand that these comments may require additional effort on your part. However, addressing these issues will improve the quality and impact of your manuscript. I look forward to seeing an improved version of your manuscript in the future.

Round 2
Reviewer 1 Report
Comments and Suggestions for Authors
Very good revision.
Author Response
Comment 1 - Very good revision.
Reply to Comment 1 - We really thank the reviewer for this reply.
Reviewer 2 Report
Comments and Suggestions for Authors
Now the novelty of the paper is a bit clearer but not enough. The authors need to explain which differences will make using old WHO and new WHO values. They need to include further description of previous research in other cities where similar methods have been used and why they were limited such studies and demonstrate that their study is better.
Comments on the Quality of English LanguageEnglish has been revised but still series errors in English even in headings.
Author Response
Now the novelty of the paper is a bit clearer but not enough. The authors need to explain which differences will make using old WHO and new WHO values. They need to include further description of previous research in other cities where similar methods have been used and why they were limited such studies and demonstrate that their study is better.
Most studies still use the values suggested by WHO in 2005 (2005-AQG), considering one or more pollutants and applying the IEHIA tool only for mortality from natural causes. The studies using the 2021-AQG are conducted either on a single Italian city evaluating health impact due to PM2.5 exposure or at a global level considering only NO2 and only mortality from natural causes. Since the existing risk functions, fundamental for the application of the IEHIA tool, are available only for a few pollutants and for limited adverse outcomes, the available scientific literature is rather scarce and heterogeneous. Furthermore, the 2021-AQG, more stringent than the 2005-AQG, are values determined by scientific evidence, representing those minimum values above which significant adverse effects on human health are documented. Therefore, carrying out the analyses using the 2021-AQG allows us to know the maximum number of deaths attributable, and therefore avoidable, to air pollution exposure. If the 2005-AQG are instead used, the number of deaths attributable to air pollution exposure would be underestimated. The novelty of this study lies in using the latest values indicated by the WHO as counterfactual, estimating therefore the overall number of avoidable deaths caused by air pollution exposure, and considering all the pollutants and all the risk functions available in the literature, updated by the WHO on the occasion of the publication of the latest guidelines, thus providing a global picture of the impact of air pollution in the city of Pisa, in terms of attributable deaths. Furthermore, a further novelty of this study lies in the use of this methodology (scientifically consolidated), commissioned for the first time at national level by the municipality itself, as a tool supporting policy makers in identifying strategies to improve urban environments and public health.
These aspects have been added into the text, in the Introduction section (lines 87-109).

Reviewer 3 Report
Comments and Suggestions for Authors
Dear authors, thank you for making corrections to the body of the manuscript. I believe that the work you have done has helped to improve the article. Thank you for conducting a comparative analysis to determine the level of atmospheric pollution in Italy. However, I believe that it is necessary to make a comparison not only with the study region, but also with other atmospheric observation centers.
Author Response
Dear authors, thank you for making corrections to the body of the manuscript. I believe that the work you have done has helped to improve the article. Thank you for conducting a comparative analysis to determine the level of atmospheric pollution in Italy. However, I believe that it is necessary to make a comparison not only with the study region, but also with other atmospheric observation centers.
A table (Table 2) has been added showing the comparisons between the values of the concentration distributions of the various pollutants taken into consideration between Pisa and the Tuscany region, the Po Valley and Italy. A comment has been added to this table (lines 163-183).